# Bayesian machine learning enables discovery of risk factors for hepatosplenic multimorbidity related to schistosomiasis

Yin-Cong Zhi[1], Victor Anguajibi[2], John B. Oryema[3], Betty Nabatte[4], Christopher K. Opio[5], Narcis B. Kabatereine[3] & Goylette F. Chami [1] ✉

One in 25 deaths worldwide is related to liver disease, and often with multiple hepatosplenic conditions. Yet, little is understood of the risk factors for hepatosplenic multimorbidity, especially in the context of chronic infections. We present a novel Bayesian multitask learning framework to jointly model 45 hepatosplenic conditions assessed using point-of-care B-mode ultrasound for 3155 individuals aged 5-91 years within the SchistoTrack cohort across rural Uganda, where chronic intestinal schistosomiasis is endemic. We identify distinct and shared biomedical, socioeconomic, and spatial risk factors for individual conditions and hepatosplenic multimorbidity, and introduce methods for measuring condition dependencies as risk factors. Notably, for gastro-oesophageal varices, we discover key risk factors of older age, lower haemoglobin concentration, and schistosomal periportal fibrosis. Our findings provide a compendium of risk factors to inform surveillance, triage, and follow-up, while our model enables improved prediction of hepatosplenic multimorbidity, and if validated on other anatomical systems, general multimorbidity.

Multimorbidity is the co-occurrence of two or more chronic conditions and is of growing global concern in existing health systems due to the lack of clinical guidelines, strains on resources and staff, and potential mismanagement or misdiagnosis of patients[1–3]. Individuals with multimorbidity are among the most socially and economically disadvantaged, with greater years lived with disability and higher mortality compared to those without multimorbidity[2]. In low and middle-income countries (LMICs), over one-third of individuals are affected by complex multimorbidity due to both infectious and noncommunicable causes[4,5]. Yet, there is little understanding of multimorbidity due to chronic infections in LMICs, with most multimorbidity studies focusing on ageing or cardiovascular diseases in high-income countries[6–8].

Schistosomiasis, caused by the intestinal species *Schistosoma mansoni*, is a major cause of hepatosplenic multimorbidity in sub-

Saharan Africa. The most common hepatosplenic conditions that occur in schistosomiasis-endemic populations include periportal fibrosis, early cirrhosis-like or hepatitis-like livers, splenomegaly, moderately shrunken or enlarged livers, and thickened gall bladder walls–all of which are often accompanied by clinical symptoms such as diarrhoea, abdominal pain, anaemia, jaundice, and melena[9–11]. Hepatosplenic multimorbidity due to chronic schistosome infections encompasses two or more interacting or co-occurring conditions ranging from combinations of less severe forms of periportal fibrosis and changes in spleen or liver organometry to inclusion of life-threatening complications of gastro-oesophageal varices and liver cirrhosis[10]. More than one in every two individuals aged five years and older living in areas endemic with *S. mansoni* have been estimated to have hepatosplenic multimorbidity[10]. Yet, there is a limited understanding of the risk factors of hepatosplenic multimorbidity or, more

[1]Big Data Institute, Nuffield Department of Population Health, University of Oxford, Oxford, UK. [2]Uganda Institute of Allied Health Sciences, Kampala, Uganda. [3]Pakwach Local District Government, Uganda Ministry of Health, Pakwach Town, Uganda. [4]Division of Vector-Borne and Neglected Tropical Diseases Control, Uganda Ministry of Health, Kampala, Uganda. [5]Aga Khan University Hospital, Nairobi, Kenya. ✉e-mail: goylette.chami@ndph.ox.ac.uk

generally, diverse hepatosplenic outcomes beyond periportal fibrosis[9]. Currently, only simple composite outcomes are studied where diverse conditions are collapsed into one indicator (periportal fibrosis[9]) or less severe conditions (diarrhoea, etc.)[12] are assessed as independent outcomes. From these singular outcome studies, current schistosome infection has been shown to be irrelevant for periportal fibrosis prediction[9], despite still being used as a proxy indicator for morbidity by the World Health Organization (WHO)[13–15]. In contrast, co-infection with human immunodeficiency virus (HIV) and hepatitis B (HBV) infections has been shown to confer over two times higher odds of periportal fibrosis coupled with the past history of schistosome exposure[9]. The complex interplay of pathogenesis provides support for a renewed focus on multimorbidity and interacting conditions. It is unknown whether co-infections result in biologically related conditions that interact to exacerbate schistosomiasis-specific conditions or cause independent co-occurring conditions due to shared socio-economic and spatial risk factors. Life-threatening conditions related to schistosomiasis, such as gastro-oesophageal varices that are indicative of portal hypertension, remain to be studied outside of health facilities and beyond basic demographic risk factors[11], and little is known about how multimorbidity influences schistosomiasis-related condition progression.

For schistosomiasis, the common approach of focusing narrowly on one condition at a time fails to account for all conditions currently relevant to a patient. Consequently, there is a limited understanding of risk factors shared across conditions and, critically, condition inter-dependencies, to be able to identify early indicators of severe conditions and disease progression. This oversimplified approach to multimorbidity modelling is not unique to schistosomiasis and applies more generally to recent attempts to study multimorbidity. Many studies simply identify multimorbidity as a binary indicator of two or more conditions from an arbitrary list of predefined conditions, regardless of the actual condition type or interaction[2,5,16,17].

Given multimorbidity arises from shared risk factors including condition inter-dependencies, we developed a Bayesian multitask learning architecture[18,19], allowing each condition to learn from the risk factors of other conditions[20]. Graphs (or networks) were utilised to encode inter-dependencies between conditions, and parameterised to homogenise the predictions based on node connectivity. We jointly predicted 45 hepatosplenic conditions assessed through point-of-care B-mode ultrasound (with the WHO Niamey Protocol for defining different periportal fibrosis patterns[21]) for 3155 individuals aged 5–91 years within the community-based cohort, SchistoTrack, in rural Uganda. The aim of this study was to employ Bayesian machine learning and multitask modelling to discover shared risk factors from a wide range of biomedical, sociodemographic, and spatial variables for hepatosplenic multimorbidity, in particular life-threatening severe hepatosplenic conditions, and to measure the influence of condition dependencies.

## Results

### Risk factors for hepatosplenic conditions
An illustration of how the results were obtained can be found in Fig. 1. In our study population, 82% (2578/3155) of individuals had at least one condition, and 54% (1708/3155) had two or more conditions. When considering the maximum count of multimorbidity across all study participants including those without multimorbidity, the top 5.9% (185/3155) of all study participants had six or more conditions (maximum number of conditions in any one individual was 13); the top 5.9% across all study participants corresponded to the top 10.8% of multimorbid participants (185/1708). Mild liver fibrosis (Niamey protocol patterns C1 and C2), liver shrunkenness, and spleen enlargement were the most commonly observed conditions (see Table 1 for prevalence of all conditions). By examining a wide set of participant covariates (Table 2), we first identified significant risk factors for each of the individual 45 conditions (Table 3).

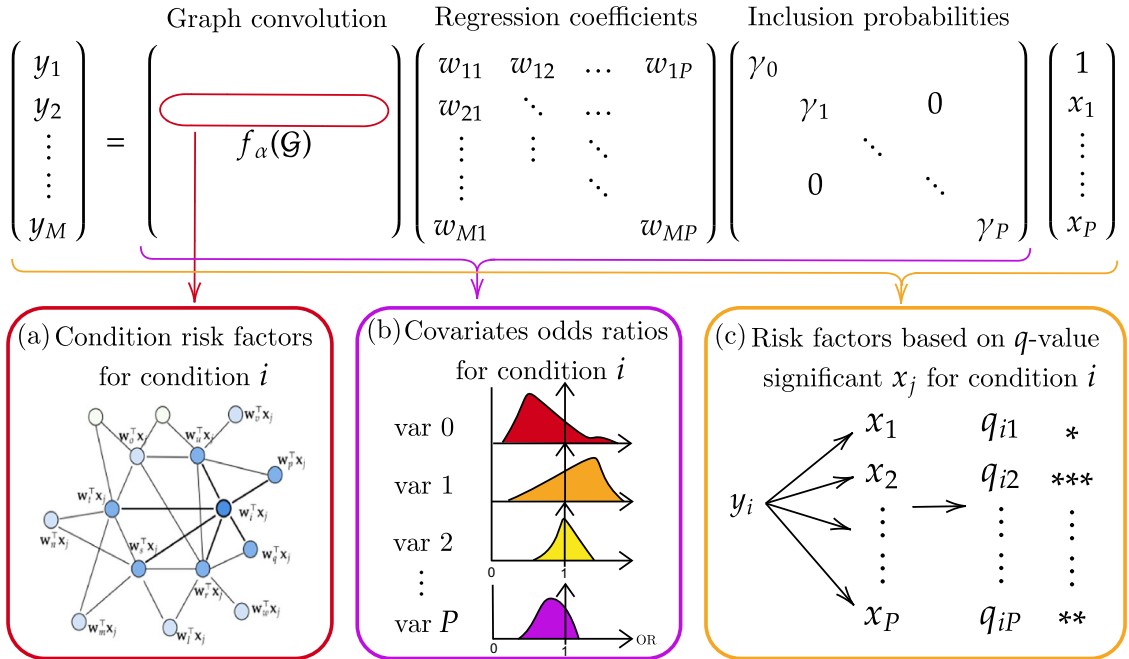

**Fig. 1 | Pipeline of study.** Each panel depicts an interpretable element of the model: **a** Identify the strongest dependencies to node $i$ based on the graph convolution weights to inform the conditions that have the strongest influence on condition $i$. These contribute to the list of risk factors of condition $i$. the weights can also be visualised on the graph. **b** Determine the effect size of covariates by examining posterior odds ratios, with reverse standardisation applied to match the original units. Median and 95% HPD CI intervals are computed for point estimates. **c** Measure the significance of covariates against all outcomes by likelihood ratio tests, with 5% significance determined from $q$-values from correcting $p$-values to adjust for the false positive rate.

## Table 1 | List of outcome conditions

| Outcome group | Outcome | # Participants (%) | Severity |
|---|---|---|---|
| Liver patterns | | | |
| | *Normal* | 2312 (73.28) | None |
| | Unclear (B) | 15 (0.48) | Mild |
| | Feather streaks (B0) | 238 (7.54) | Mild |
| | Flying saucers, starry sky (B1) | 194 (6.15) | Mild |
| | Spider thickening (B2) | 53 (1.68) | Mild |
| | Prominent peripheral rings (C1) | 461 (14.61) | Mild |
| | Prominent pipe stems (C2) | 455 (14.41) | Moderate |
| | Ruff, portal bifurcation (D) | 185 (5.86) | Severe |
| | Patches (occluded, bright white vessels) (E) | 45 (1.43) | Severe |
| | Bird's claw (F) | 8 (0.25) | Severe |
| Other abnormalities | | | |
| | *None* | 2922 (92.61) | None |
| | Cirrhosis-like liver | 12 (0.38) | Severe |
| | Fatty-like liver | 64 (2.03) | Mild |
| | Chronic hepatitis or early cirrhosis | 65 (2.06) | Moderate |
| | Polycystic kidney disease | 2 (0.06) | None |
| | Liver cysts | 4 (0.13) | None |
| | Situs inversus | 1 (0.03) | None |
| | Other | 47 (1.49) | Unclear |
| Liver surface | | | |
| | *None* | 3104 (98.38) | None |
| | Slight/serrated | 17 (0.54) | Moderate |
| | Gross/undulating | 34 (1.08) | Severe |
| Caudal liver edge | | | |
| | *Sharp* | 2881 (91.32) | None |
| | Rounded | 274 (8.68) | Mild |
| Left liver lobe | | | |
| | *Normal* | 2189 (69.38) | None |
| | Moderately enlarged | 422 (13.38) | Mild |
| | Moderately shrunken | 357 (11.32) | Moderate |
| | Severely enlarged | 84 (2.66) | Mild |
| | Severely shrunken | 103 (3.26) | Moderate |
| Right liver lobe | | | |
| | *Normal* | 2179 (69.06) | None |
| | Moderately enlarged | 404 (12.81) | Mild |
| | Moderately shrunken | 409 (12.96) | Moderate |
| | Severely enlarged | 51 (1.62) | Mild |
| | Severely shrunken | 112 (3.55) | Severe |
| Mean portal vein | | | |
| | *Normal* | 2138 (67.77) | None |
| | Moderately enlarged | 421 (13.34) | Moderate |
| | Moderately restricted | 382 (12.11) | Unclear |
| | Severely enlarged | 166 (5.26) | Mild |
| | Severely restricted | 48 (1.52) | Unclear |
| Portosystemic collaterals | | | |
| | *Not detected* | 3121 (98.92) | None |
| | Splenic varices | 12 (0.38) | Severe |

## Table 1 (continued) | List of outcome conditions

| Outcome group | Outcome | # Participants (%) | Severity |
|---|---|---|---|
| | Gastro-oesophageal varices | 11 (0.35) | Severe |
| | Pancreaticoduodenal varices | 5 (0.16) | Severe |
| | Entirely recanalised para-umbilical vein ≥3mm | 1 (0.03) | Severe |
| | Splenorenal shunt | 13 (0.41) | Severe |
| | Other | 1 (0.03) | Severe |
| Ascites | | | |
| | *Not detected* | 3143 (99.62) | None |
| | Yes | 12 (0.38) | Moderate |
| Gall bladder visible | | | |
| | Not visible | 30 (0.95) | Severe |
| | Yes, but blocked by a stone or collapsed | 115 (3.65) | None |
| | *Yes, clearly visible* | 3010 (95.40) | None |
| Gall bladder wall | | | |
| | *Normal* | 2908 (92.17) | None |
| | Thick | 247 (7.83) | Severe |
| Spleen length | | | |
| | *Normal* | 2164 (68.59) | None |
| | Moderately enlarged | 384 (12.17) | Moderate |
| | Moderately shrunken | 390 (12.36) | None |
| | Severely enlarged | 180 (5.71) | Severe |
| | Severely shrunken | 37 (1.17) | None |

Condition groupings are indicated by the first column. Participants can have multiple conditions from the same group except for organ measurements (left and right liver lobes, mean portal vein, and spleen length), which are mutually exclusive. In each group, the healthy status is italicised and used as a reference. For liver patterns, Niamey protocol grades are indicated in brackets. Summary is over 3155 participants, prevalence across all conditions is 0.046.

Risk factors significant for at least one condition included three clinical measurements, seven sociodemographic variables, and three spatial factors. Haemoglobin (Hb) concentration was the most informative measurement and was negatively related to 24.44% (11/45) of conditions (natural log-transformed average median odds ratios (OR) 0.20, range 0.03–0.54); among associated conditions, four were severe, four moderate, and three mild with respect to portal hypertension risk. Notably, among these conditions, significant variation in WHO categories of anaemia status was observed for chronic hepatitis or early cirrhosis, gastro-oesophageal varices, splenorenal shunts, ascites, and severely enlarged spleens (Supplementary Table S1). Malaria infection and *S. mansoni* eggs per gram (EPG) were associated with few non-severe conditions (three and two, respectively). Age was relevant for most conditions (64.44%, 29/45), irrespective of severity (10 severe, four moderate, 12 mild), with an average median OR of 1.03 (range 0.97–1.06), and almost all significant conditions were positively related to older age (89.66%, 26/29). Both gender (female) and being a fisherman were related to six conditions (13.33%). All associations with being female were negative (average median OR 0.64, range 0.53–0.72), including two severe conditions, and all associations with being a fisherman were positive (average median OR 1.91, range 1.51–2.13), including three severe conditions. Access to an improved drinking water source had two negative (median OR range 0.74–0.75) and one positive (median OR 1.57) association with only non-severe conditions. The number of individuals in the household (positive), belonging to the majority tribe (negative), and belonging to a home that was owned (i.e. not rented) (positive) were related to one non-severe condition each (OR range 0.66–1.53). Spatial risk factors

## Table 2 | List of covariates

| Covariate | Category | Summary type | Summary |
|---|---|---|---|
| *S. mansoni* eggs per gram (log(+1)) | | Mean (SD) | 1.40 (2.19) |
| *S. mansoni* eggs per gram (raw) | | Mean (SD) | 85.25 (404.28) |
| Malaria rapid diagnostic test | | Count (%) | 1469 (46.56) |
| HBV | | Count (%) | 179 (5.67) |
| HIV (self-reported) | | Count (%) | 55 (1.74) |
| Haemoglobin concentration (log) | | Mean (SD) | 2.51 (0.22) |
| Haemoglobin concentration (raw, g/dL) | | Mean (SD) | 12.62 (2.53) |
| Age | | Mean (SD) | 25.93 (18.39) |
| Gender | *Male* | Count (%) | 1457 (46.18) |
| | Female | Count (%) | 1698 (53.82) |
| Majority tribe | | Count (%) | 2404 (76.20) |
| Majority religion | | Count (%) | 2420 (76.70) |
| Years in education | | Mean (SD) | 3.25 (2.78) |
| Occupation | *Other/None* | Count (%) | 2296 (72.77) |
| | Farmer | Count (%) | 540 (17.12) |
| | Fisherman | Count (%) | 219 (6.94) |
| | Fishmonger | Count (%) | 100 (3.17) |
| Home quality score | | Mean (SD) | 5.60 (3.41) |
| Household social status | | Count (%) | 385 (12.20) |
| Number of individuals in household | | Mean (SD) | 3.65 (1.42) |
| Years the household lived in the village | | Mean (SD) | 20.10 (15.85) |
| Home owned | | Count (%) | 2759 (87.45) |
| Number of rooms | | Mean (SD) | 2.14 (1.19) |
| Current alcohol use | | Count (%) | 306 (9.70) |
| Improved drinking water source | | Count (%) | 1748 (55.40) |
| Number of water activities | | Mean (SD) | 1.76 (0.97) |
| Year of recruitment | *2022* | Count (%) | 2207 (69.95) |
| | 2023 | Count (%) | 948 (30.05) |
| Min. dist. (km) to water site | | Mean (SD) | 0.52 (0.44) |
| Min. dist. (km) to health centre | | Mean (SD) | 4.11 (3.31) |
| District | *Mayuge* | Count (%) | 754 (23.90) |
| | Buliisa | Count (%) | 1006 (31.89) |
| | Pakwach | Count (%) | 1395 (44.22) |

Reference categories of categorical variables are italicised. Summaries are calculated from a total of 3155 participants. When log transforms are taken, summaries of the raw values are also provided but are not used for modelling.

## Table 3 | List of significant relationships found between all conditions and covariates

| Condition | Covariate | Median OR (95% CI) |
|---|---|---|
| Liver pattern unclear | Age | 1.04 (1.02, 1.06) |
| Feather streaks | Age | 0.98 (0.97, 0.99) |
| Feather streaks | Pakwach | 1.43 (0.98, 2.05) |
| Flying saucers | Malaria | 1.46 (1.07, 2.06) |
| Flying saucers | Age | 0.97 (0.96, 0.99) |
| Flying saucers | Pakwach | 1.63 (1.14, 2.55) |
| Spider thickening | Age | 1.03 (1.01, 1.04) |
| Prominent peripheral rings | Age | 1.03 (1.02, 1.04) |
| Prominent peripheral rings | Gender - female | 0.62 (0.49, 0.79) |
| Prominent peripheral rings | Fisherman | 2.02 (1.48, 2.81) |
| Prominent peripheral rings | Number of individuals in HH | 1.09 (1.03, 1.16) |
| Prominent peripheral rings | Buliisa | 1.48 (1.06, 2.05) |
| Prominent peripheral rings | Pakwach | 2.21 (1.55, 3.03) |
| Prominent peripheral rings | Improved drinking water source | 0.76 (0.62, 0.89) |
| Prominent pipe stems | Age | 1.03 (1.02, 1.03) |
| Prominent pipe stems | Gender - female | 0.65 (0.52, 0.77) |
| Prominent pipe stems | Fisherman | 2.10 (1.55, 3.02) |
| Prominent pipe stems | Buliisa | 1.41 (1.07, 1.95) |
| Prominent pipe stems | Pakwach | 2.06 (1.52, 2.73) |
| Prominent pipe stems | Improved drinking water source | 0.74 (0.61, 0.89) |
| Ruff portal bifurcation | Age | 1.06 (1.05, 1.07) |
| Ruff portal bifurcation | Gender - female | 0.54 (0.36, 0.81) |
| Ruff portal bifurcation | Fisherman | 2.13 (1.34, 3.36) |
| Patches | log(Hb concentration) | 0.20 (0.08, 0.58) |
| Patches | Age | 1.05 (1.03, 1.06) |
| Birds claw | Age | 1.04 (1.02, 1.07) |
| Cirrhosis-like liver | Age | 1.06 (1.03, 1.09) |
| Fatty-like liver | Age | 1.04 (1.02, 1.05) |
| Chronic hepatitis or early cirrhosis | log(Hb concentration) | 0.15 (0.06, 0.38) |
| Chronic hepatitis or early cirrhosis | Age | 1.03 (1.01, 1.04) |
| Chronic hepatitis or early cirrhosis | Pakwach | 2.58 (1.30, 5.15) |
| Other abnormalities | Age | 1.04 (1.03, 1.06) |
| Right liver lobe moderately enlarged | Age | 1.01 (1.01, 1.02) |
| Right liver lobe moderately enlarged | Buliisa | 1.46 (1.08, 1.93) |
| Right liver lobe moderately shrunken | Buliisa | 0.73 (0.52, 0.95) |
| Right liver lobe severely enlarged | Age | 1.04 (1.02, 1.05) |
| Left liver lobe moderately enlarged | log(Hb concentration) | 0.53 (0.34, 0.81) |
| Left liver lobe moderately enlarged | Age | 1.02 (1.01, 1.02) |
| Left liver lobe moderately enlarged | Buliisa | 1.84 (1.30, 2.77) |
| Left liver lobe moderately enlarged | Pakwach | 1.94 (1.40, 2.87) |
| Left liver lobe moderately shrunken | Buliisa | 0.64 (0.47, 0.85) |
| Left liver lobe moderately shrunken | Pakwach | 0.74 (0.55, 0.96) |

included living in the Western region of Uganda and distance to open freshwater sites. Pakwach district was associated to 31.11% (14/45, 12/14 positive) of conditions, and Buliisa district to 17.78% (8/45, 6/8 positive) when compared to Mayuge district, of which only three and one condition, respectively, were severe. The minimum distance (km) to an open freshwater site was positively associated with only one mild condition. The OR for every condition is presented in Supplementary Fig. S1, with the mean to show the directional effect.

Gastro-oesophageal varices were the primary focus of our analysis because they indicate severe portal hypertension. All covariate significance values for the condition are shown in Supplementary

**Table 3 (continued) | List of significant relationships found between all conditions and covariates**

| Condition | Covariate | Median OR (95% CI) |
|---|---|---|
| Left liver lobe severely enlarged | log(Hb concentration) | 0.23 (0.09, 0.51) |
| Left liver lobe severely enlarged | Age | 1.04 (1.02, 1.05) |
| Liver surface gross undulating | Age | 1.04 (1.02, 1.06) |
| Liver surface gross undulating | Pakwach | 3.94 (1.57, 11.63) |
| Caudal liver edge rounded | *S. mansoni* log(EPG+1) | 1.08 (1.01, 1.14) |
| Caudal liver edge rounded | log(Hb concentration) | 0.25 (0.15, 0.47) |
| Caudal liver edge rounded | Age | 1.01 (1.00, 1.02) |
| Caudal liver edge rounded | Buliisa | 1.94 (1.34, 3.01) |
| Caudal liver edge rounded | Pakwach | 1.39 (1.06, 1.95) |
| Caudal liver edge rounded | Improved drinking water source | 1.51 (1.16, 1.98) |
| Caudal liver edge rounded | Min. dist. (km) to water site | 2.10 (1.50, 2.89) |
| Mean portal vein moderately enlarged | *S. mansoni* log(EPG+1) | 1.07 (1.02, 1.13) |
| Mean portal vein moderately enlarged | Age | 1.02 (1.01, 1.03) |
| Mean portal vein moderately enlarged | Fisherman | 1.52 (1.07, 2.22) |
| Mean portal vein moderately restricted | Age | 0.98 (0.97, 0.98) |
| Mean portal vein severely enlarged | log(Hb concentration) | 0.18 (0.09, 0.35) |
| Mean portal vein severely enlarged | Age | 1.05 (1.04, 1.06) |
| Mean portal vein severely enlarged | Pakwach | 1.94 (1.35, 3.25) |
| Mean portal vein severely restricted | Majority tribe | 0.68 (0.38, 1.05) |
| Splenic varices | Age | 1.04 (1.01, 1.07) |
| Gastro-oesophageal varices | log(Hb concentration) | 0.07 (0.01, 0.37) |
| Gastro-oesophageal varices | Age | 1.04 (1.02, 1.08) |
| Splenorenal shunt | log(Hb concentration) | 0.08 (0.02, 0.42) |
| Splenorenal shunt | Age | 1.03 (1.02, 1.06) |
| Ascites | log(Hb concentration) | 0.03 (0.01, 0.17) |
| Ascites | Age | 1.04 (1.02, 1.07) |
| Gall bladder not visible | Buliisa | 4.58 (1.46, 11.76) |
| Gall bladder blocked by a stone/collapsed | Gender - female | 0.59 (0.39, 0.95) |
| Gall bladder wall thick | Age | 1.03 (1.02, 1.03) |
| Gall bladder wall thick | Gender - female | 0.71 (0.53, 0.90) |
| Gall bladder wall thick | Fisherman | 1.59 (1.16, 2.39) |
| Gall bladder wall thick | Pakwach | 1.52 (1.12, 2.22) |
| Spleen length severely enlarged | log(Hb concentration) | 0.07 (0.03, 0.13) |
| Spleen length severely enlarged | Age | 1.03 (1.02, 1.04) |
| Spleen length severely enlarged | Fisherman | 2.01 (1.28, 3.40) |
| Spleen length severely enlarged | Pakwach | 2.63 (1.51, 4.15) |
| Spleen length severely shrunken | Age | 1.02 (1.01, 1.04) |
| Spleen length moderately enlarged | Malaria | 1.68 (1.31, 2.08) |

**Table 3 (continued) | List of significant relationships found between all conditions and covariates**

| Condition | Covariate | Median OR (95% CI) |
|---|---|---|
| Spleen length moderately enlarged | log(Hb concentration) | 0.32 (0.20, 0.51) |
| Spleen length moderately enlarged | Home owned | 1.54 (1.00, 2.41) |
| Spleen length moderately enlarged | Pakwach | 1.78 (1.26, 2.44) |
| Spleen length moderately shrunken | Malaria | 0.37 (0.29, 0.50) |
| Spleen length moderately shrunken | Gender - female | 0.72 (0.55, 0.89) |
| Spleen length moderately shrunken | Pakwach | 0.54 (0.39, 0.80) |

Outcome conditions and their significant covariates are listed in the first two columns; significance is calculated to 5% based on corrected *q*-values. Results are computed over the full dataset.

Table S2; the corresponding posterior ORs are visualised in Supplementary Fig. S2. Age and Hb concentration were the only significant variables. Each additional year of age was associated with a 4% increase in the odds of gastro-oesophageal varices (median OR 1.04, 95% credible interval (CI) 1.02–1.08). A 10% increase in the log-transformed Hb concentration corresponded to a 22% decrease in the likelihood of gastro-oesophageal varices (OR 0.07, 95% CI 0.01–0.37). From the covariate densities, a number of clear moderate to large non-zero effects of other risk factors were observed, but were borderline insignificant from uncorrected *p*-values. HIV (OR 2.52, CI 0.97–10.77), HBV (OR 1.40, CI 0.81–3.72), female (OR 0.67, CI 0.26–1.04), and fisherman (OR 1.79, CI 0.91–4.44) had credible intervals narrowly including 1. The non-zero effect was visible through the heavy tails of skewed posterior densities, while insignificance was due to the modal OR remaining close to one. In sensitivity analyses, we investigated the models with alternative priors as discussed in the "Methods" section. The posterior ORs for gastro-oesophageal varices from the alternative priors are shown in Supplementary Fig. S3. The inference was robust to prior choice as long as they remained theoretically appropriate for the model components, as the OR distributions were extremely similar to Supplementary Fig. S2.

Group risk factors were computed to profile periportal fibrosis associated with schistosomiasis through combined likelihood ratios of the five liver fibrosis patterns (Niamey protocol patterns C–F as defined by the features adjacent to portal vasculature) (Supplementary Table S2). The direction of association of each covariate was determined by the mean from combining the five posterior OR densities. Periportal fibrosis shared many risk factors with the five patterns individually, with the addition of HIV, despite being insignificant alone for the singular liver fibrosis patterns. HIV exhibited a positive mean effect, with infection leading to a 1.77 times higher odds of developing periportal fibrosis (Niamey protocol patterns C–F). Other significant associations were found with older age (mean OR 1.04), being female (mean OR 0.62), being a fisherman (mean OR 2.00), the number of individuals in the household (mean OR 1.09), access to improved drinking water source (mean OR 0.95), and the Western districts (Buliisa mean OR 1.31, Pakwach mean OR 1.99).

Significance against overall hepatosplenic multimorbidity was also computed on the 45 conditions (Supplementary Table S2). Predicting overall multimorbidity represented the expected morbidity counts in each participant; scenarios that lead to the highest overall multimorbidity would represent the expected maximum morbidity count. Six risk factors previously observed to be significant for individual conditions remained. The factors leading to higher expected morbidity counts included lower Hb concentration, older age, being

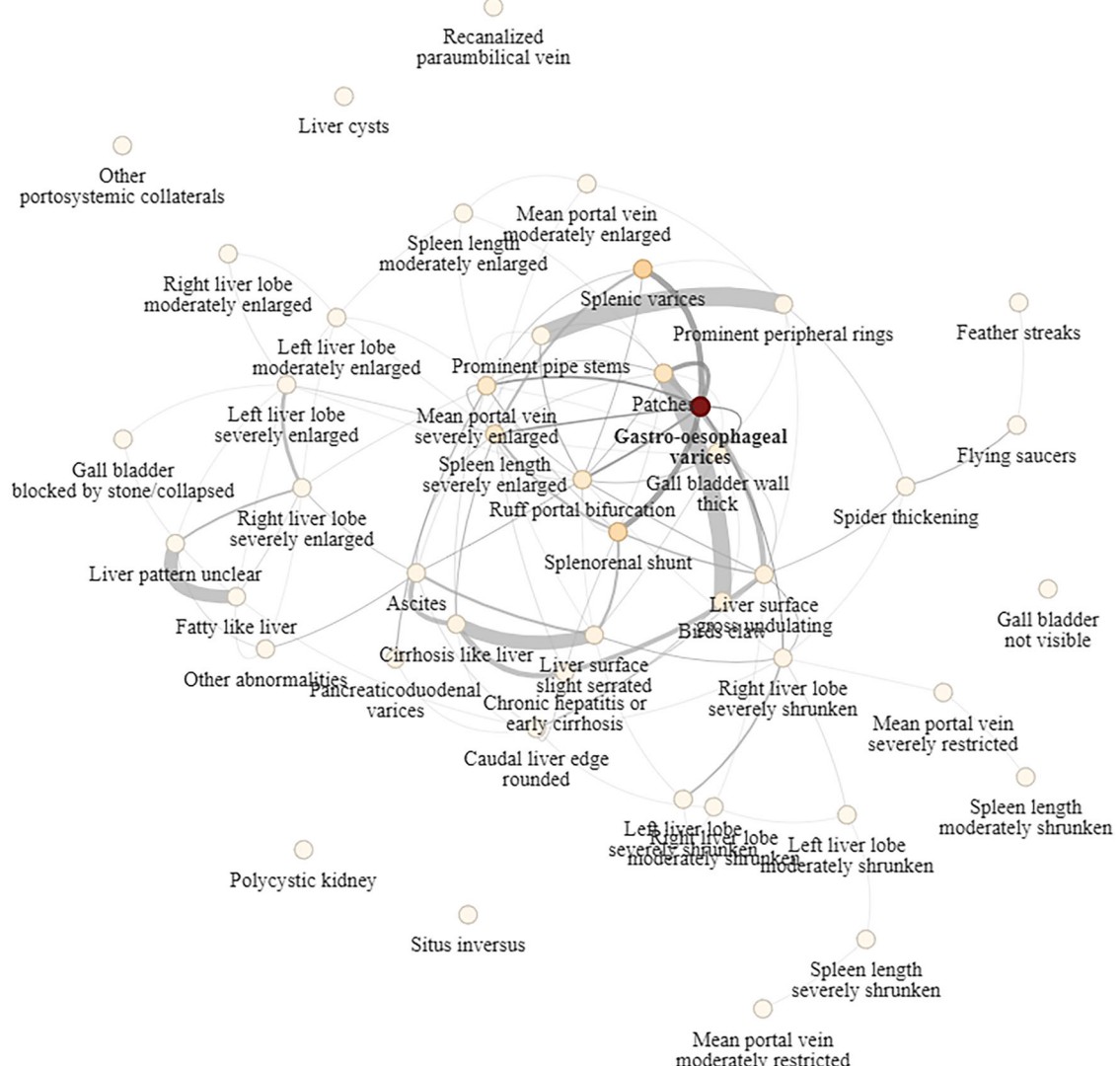

**Fig. 2 | Influence probabilities on predicting gastro-oesophageal varices from models trained on other conditions.** Probability on each node determines the proportion of influence from the model trained on the node condition on predicting gastro-oesophageal varices; darker colour implies larger influence, lighter colour implies minor influence. Probabilities are obtained from the row corresponding to gastro-oesophageal varices in the graph convolution matrix $f_\alpha(\mathcal{G})$, where $\alpha$ is chosen as the median of its posterior (see Supplementary Fig. S4). The exact probabilities of the conditions can be found in Supplementary Fig. S19.

male, working as a fisherman, and living in the Western districts (with the strongest effect in Pakwach district). To identify conditions with the highest probabilities for overall hepatosplenic multimorbidity, an extreme scenario was simulated using the top 99% percentile or the bottom 1% percentile of the significant covariate values (depending on the direction of association) while keeping insignificant covariates at the median value (Supplementary Fig. S23). The expected number of conditions the patient will have in this extreme scenario is 7.86, while the most likely conditions to develop were severe enlargement of the spleen (probability 0.69), and any form of enlargement of the portal vein (probability 0.69 for moderate enlargement and 0.64 for severe enlargement). Findings for individual conditions, periportal fibrosis, and overall hepatosplenic multimorbidity remained robust when stratified by children (aged 5-17 years) and adults (aged 18+ years), notably revealing that most risk factor associations were found in adults (Supplementary Tables S3–S6).

## Graph convolution and condition inter-dependencies

Condition dependencies for gastro-oesophageal varices using the graph are shown in Fig. 2, where node weightings (condition dependency measure) for the condition were controlled by parameter $\alpha$ that was chosen as the median value from its posterior MCMC samples (Supplementary Fig. S4). A set of probabilities between all conditions can be found in Supplementary Fig. S5. The self-influence for gastro-oesophageal varices indicated a probability of 0.397, measuring how much was predicted from its own model and risk factors/covariates. From the other condition models, the highest probability contributions to gastro-oesophageal varices came from related biological complications of splenic varices and splenorenal shunting, conferring additional probabilities of 0.102 and 0.081, respectively. The next most relevant conditions encompassed potential fibrotic precursors to portal hypertension and included moderate and severe forms of schistosomal periportal fibrosis. The rough liver pattern (Niamey protocol pattern D), where there was extensive fibrosis at the point of portal bifurcation, increased the probability by 0.045. Liver patches (Niamey protocol pattern E), which were indicative of occluded vessels and fibrosis blocking the main portal vein, conferred an increased probability of 0.059. Other condition dependencies included a severely enlarged spleen (0.049 probability) and a severely enlarged (mean) portal vein diameter (0.046 probability). All other

**Table 4 | Model performance comparison**

| Model | AUC | AUC-PR | F1 (Bin) | Prec (Bin) | Rec (Bin) | PPV (90%) |
|---|---|---|---|---|---|---|
| LR (AIC) | 0.636 ± 0.007 | 0.089 ± 0.003 | 0.236 ± 0.006 | 0.164 ± 0.007 | 0.422 ± 0.039 | 0.103 ± 0.002 |
| XGBoost | 0.671 ± 0.011 | 0.092 ± 0.004 | 0.239 ± 0.004 | 0.163 ± 0.008 | 0.450 ± 0.050 | 0.106 ± 0.005 |
| MO-RF | 0.642 ± 0.009 | 0.096 ± 0.003 | 0.237 ± 0.003 | 0.162 ± 0.009 | 0.442 ± 0.045 | 0.114 ± 0.002 |
| ST-NN | 0.647 ± 0.014 | 0.092 ± 0.003 | 0.236 ± 0.005 | 0.164 ± 0.011 | 0.426 ± 0.055 | 0.106 ± 0.005 |
| MT-NN | 0.703 ± 0.012 | 0.099 ± 0.004 | 0.253 ± 0.004 | 0.176 ± 0.008 | 0.456 ± 0.039 | 0.115 ± 0.003 |
| BMO-LR | 0.665 ± 0.019 | 0.098 ± 0.002 | 0.236 ± 0.005 | 0.166 ± 0.005 | 0.415 ± 0.024 | 0.102 ± 0.004 |
| BMT-G | 0.694 ± 0.011 | 0.102 ± 0.004 | 0.256 ± 0.004 | 0.177 ± 0.007 | 0.461 ± 0.035 | 0.117 ± 0.006 |
| BMT-CL | 0.718 ± 0.012 | 0.101 ± 0.002 | 0.258 ± 0.004 | 0.182 ± 0.006 | 0.436 ± 0.021 | 0.122 ± 0.003 |
| BMT-G&CL | 0.721 ± 0.011 | 0.103 ± 0.004 | 0.259 ± 0.005 | 0.183 ± 0.007 | 0.449 ± 0.027 | 0.122 ± 0.003 |

For F1, precision (Prec), and recall (Rec), predicted probabilities are converted to binary using a first cut-off based on the highest F1 score, evaluated using the binary option in sklearn. Positive predicted value (PPV) is computed using a second cut-off found based on to 90% sensitivity. All metrics reported were evaluated on 10 randomly sampled test sets of 50% of the data using models trained on the other 50%.

Model abbreviations: *LR (AIC)* logistic regression AIC variable selection, *MO-RF* multi-output random forest, *ST-NN* single task neural network, *MT-NN* multitask neural network, *BMO-LR* Bayesian multi-output logistic regression, *BMT-G* Bayesian multitask with graph, *BMT-CL* Bayesian multitask with covariate learning, *BMT-G&CL* Bayesian multitask with graph and covariate learning. Metrics: *AUC* area under the ROC curve, *AUC-PR* AUC-precision-recall.

conditions had less than 0.030 probability influence, with the next most relevant being bird's claw, another severe schistosomal periportal fibrosis pattern that represents a blocked main portal vein, occluded vessels, and extensive fibrosis extending to the liver capsule (Niamey protocol pattern F) with 0.027 probability.

The graph convolution probability between splenic and gastro-oesophageal varices was among the highest donated probabilities between conditions (Supplementary Table S9), and hence among the strongest connections between any two conditions. The robustness of the probabilities around gastro-oesophageal varices under perturbation is visualised in Supplementary Figs. S20, S21, and S22 for 5%, 10%, and 20% perturbations to edges, respectively. When both addition and removal of edges were applied, the weights of reported dependencies of gastro-oesophageal varices consistently remained within the inter-quartile range of the perturbed convolution weights for all percentages. When edges were added only, the weights, being normalised, generally became reduced in proportion, but the upper-quartiles continued to contain the reported dependency weights.

For periportal fibrosis (Niamey protocol patterns C to F), all patterns had self-influence probabilities greater than 0.4 (Supplementary Figs. S14–S18). The patterns were highly interdependent and featured in their dependencies. The D grade, being the intermediate grade between assumed fibrosis progression from C1/C2 to E, was in the top five for all three dependencies, with influence probabilities from 0.023 to 0.038 and 0.030, respectively. C1 and C2 are two different cross-sections of the liver that capture similar levels of vessel-related fibrosis, and as such exhibited strong dependence (0.275–0.278 probabilities). The thickening and presentation of many second-order branches (spider thickening fibrosis, Niamey protocol pattern B2, a non-periportal fibrosis-associated liver pattern) also appeared in the top five for the two C patterns (0.018 - 0.032 probabilities). The most severe grades of schistosomal periportal fibrosis, which were anatomically nested (Niamey protocol patterns E and F), were also strongly connected (>0.175 probabilities), and strikingly, both E and F were highly dependent on a gross undulating liver surface (0.041–0.069 probabilities), typically indicative of liver cirrhosis.

To give an alternative view on condition groupings using the graph convolution network, we show the clustering of conditions found through community detection on the graph (Supplementary Fig. S24). Three main clusters were found. One cluster mainly contained measurements of liver enlargement, fatty livers, and other abnormalities not specifically attributable to infections and potentially due to alcohol use. Another cluster included cirrhosis conditions that represent features, complications, and consequences of cirrhosis that

may be attributable to alcohol use, HIV, or HBV, such as rounded liver edges, shrunken liver lobes, and gross liver surface irregularities. The third cluster encompassed mostly schistosomal periportal fibrosis Niamey patterns, its features such as changes in the main portal vein, and portal hypertension consequences, including gastro-oesophageal varices and splenorenal shunts.

## Multimorbidity prediction

The predictive performances based on a 50:50 training-testing split of the data were compared against widely used machine learning models featuring various frameworks for multiple outcomes. Models used for comparison included logistic regression, XGBoost, multi-output random forest, and shallow neural networks with single tasks and multi-task architectures. These results can be found in Table 4, with some additional metrics also presented in Supplementary Table S7. Our Bayesian multitask model using both the graph and covariate learning (BMT-G&CL), which was the focus of all aforementioned results, was superior even against models with far more complex structures, producing the highest AUC, AUC-precision-recall (AUC-PR), F1 score, precision, and positive predicted values (PPV). The reduced version, consisting of only the covariate learning multitask element (BMT-CL), performed nearly as well on all metrics and was also among the top-performing models. There was a notable difference between models that were multitasking and single-tasking (i.e. treating outcomes separately). Our Bayesian multitask models (full model and without either the graph or covariate learning component), along with the multitask neural network (MT-NN), had markedly higher AUC, F1, and PPV, while also performing marginally better with AUC-PR and precision due to the low prevalence leading to low values across all models.

We show the predictions from the BMT-G&CL for individual conditions using the AUC-PR in Supplementary Fig. S6, with the prevalence of each condition marked for reference. The AUC-PR alone indicated that the best predicted conditions were generally of higher prevalence, but we also assessed AUC-PR accounting for the prevalence, which is equivalent to comparing against random classification (which will have expected AUC-PR equal to the prevalence). This comparison was represented by the AUC-PR-to-prevalence ratio, and can be found in Fig. 3. The ratios indicated that our model generally had greater proportional improvement on rarer and often severe conditions compared to random classification. In particular, the model was over 20 times better at predicting the existence of ascites, pancreaticoduodenal varices, and recanalised paraumbilical vein, and most importantly, over 15 times better at predicting gastro-

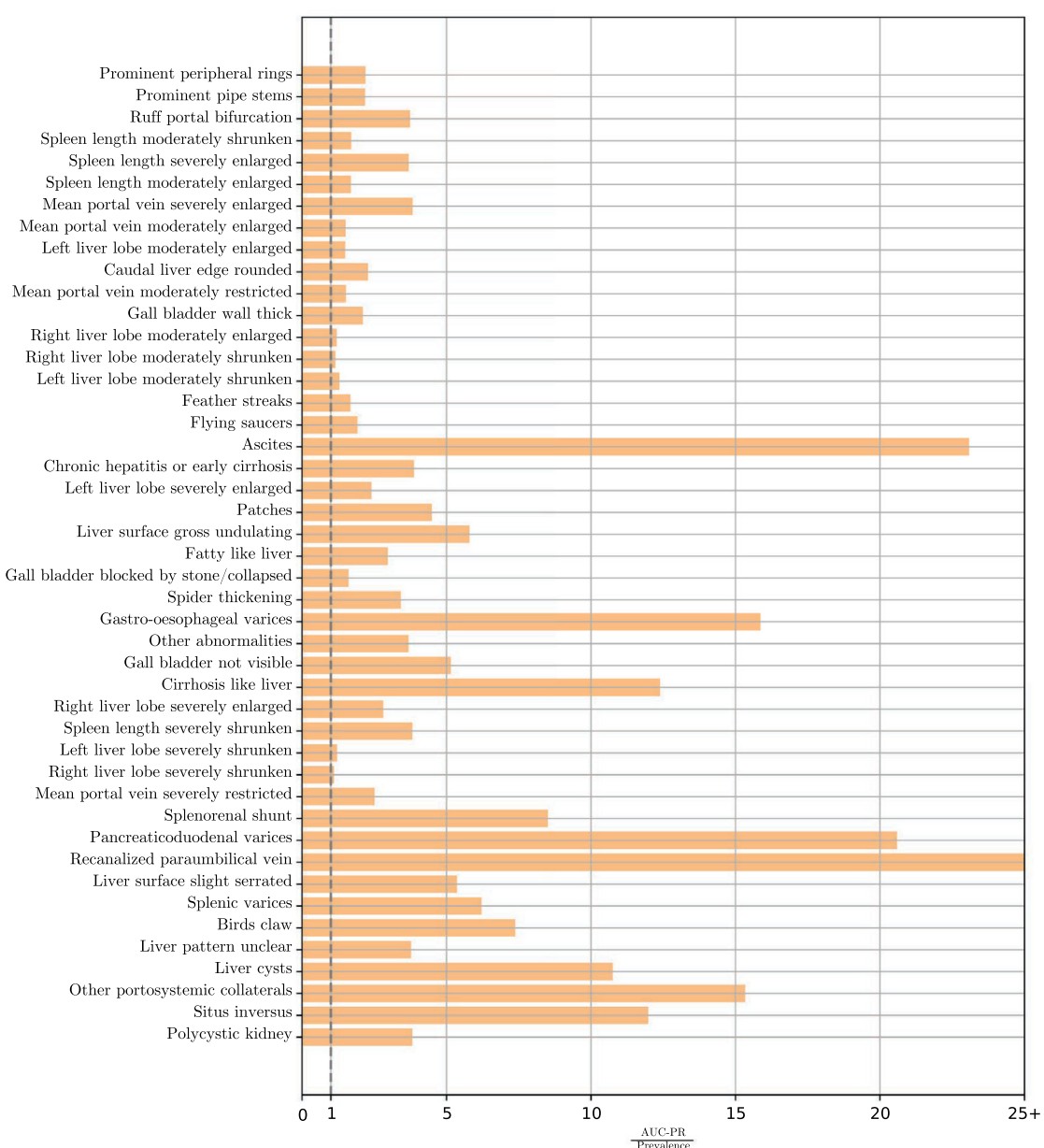

**Fig. 3 | AUC-PR-to-prevalence ratios of the Bayesian multitask model.** The ratios are calculated from our Bayesian multitask model's mean AUC-PR and prevalence values; conditions are listed in the same order as Supplementary Fig. S6. AUC-PRs were evaluated on 10 randomly sampled test sets of 50% of the data using models trained on the other 50%.

oesophageal varices and non-specified portosystemic collaterals when compared to random classification.

When comparing predictions on individual conditions, the BMT-G&CL did not show any consistent differences to the MT-NN (best performing alternative model) on any particular metric (AUC Fig. S8, AUC-PR Fig. S10, and PPV Fig. S12). Against logistic regression (the standard epidemiological approach), BMT-G&CL had superior AUC (Supplementary Fig. S7) in a number of conditions, but few significant differences were observed in AUC-PR (Fig. S9) and PPV (Fig. S11). AUC and AUC-PR curves can be found in Supplementary Fig. S13.

To understand district generalisability, a predictive experiment holding out one district at a time is presented in Supplementary Table S8. District-based prediction depended on the metric and the method used to select probability thresholds for converting to binary, but the Mayuge district was the best predicted in terms of AUC-PR despite having the lowest overall prevalence in its population.

## Discussion

Schistosomiasis-related morbidities are poorly understood, with no clinical or WHO guidelines for case management[15]. Schistosomiasis is not a single disease but a complex set of conditions (multimorbidity) arising partially due to past schistosome infection but also a wide range of diverse, often interacting causes. We studied a comprehensive list of risk factors and condition inter-dependencies for hepatosplenic multimorbidity. 3155 individuals aged 5–91 were clinically assessed for 45 liver and spleen conditions using point-of-care B-mode ultrasound within a community-based cohort (SchistoTrack[22]) in rural Uganda. Bayesian multitask learning models were developed for identifying risk factors across the 45 conditions, where we accounted for condition influences using a graph structure and shared influence probabilities. Here, we show that hepatosplenic multimorbidity is common where schistosomiasis is endemic and is associated with diverse biomedical, social, and spatial risk factors as

well as inter-dependencies between biologically interrelated conditions.

Older age was a strong, consistent positive predictor of a large number of diverse individual hepatosplenic conditions, the group of liver patterns representative of periportal fibrosis, and overall hepatosplenic multimorbidity. The finding of older age and multimorbidity is consistent with studies of widely different diseases, e.g. cardiovascular diseases[6]. However, our study now demonstrates the importance of age even in the context of chronic infections, suggesting that age is not simply an indicator of multimorbidity in LMICs due to the conventionally proposed effects of increased frailty with natural aging[5]. Our study population was relatively young compared to the life expectancy in high-income countries, with 87% of individuals less than 50 years old. However, the positive association still reflects the association of ageing and the longer time period available for acquiring multiple conditions, as shown for general multimorbidity patterns[2,16,17,23,24]. Given the high endemicity of *S. mansoni*, *Plasmodium falciparum*, HBV, and HIV, the chronic or repeated exposure to these pathogens over the life course may also have been represented through age, despite our study only measuring current infections. Future work, such as that ongoing in SchistoTrack[22], is needed to study the rate of multimorbidity accumulation over age that is attributable to chronic pathogen exposure, especially given the widespread availability of treatment and interventions for common pathogens contributing to individuals living longer with infectious diseases.

Higher Hb concentration was significantly associated with lower likelihoods of 11 hepatosplenic conditions. Most of these conditions encompassed characteristics of severe schistosomal morbidity related to portal hypertension, including gastro-oesophageal varices, but also splenorenal shunts, ascites, and splenic enlargement. The associations were robust to subgroup analyses in adults only. Hb concentration may be relevant for hepatosplenic multimorbidity due to individuals losing blood from burst gastro-oesophageal varices[11]. Alternatively, the association of Hb with hypersplenism suggests splenic dysfunction and immune impairment resulting in excessive red blood cell sequestration[25]. Chronic infection with *S. mansoni* also contributes to low Hb concentration[26]. Hb and hypersplenism were associated independently of malaria, which had no association with any splenic enlargement in adults. It is unlikely the association of Hb and hypersplenism is due to sickle cell disease, where more commonly hypersplenism is observed in children who are treated with a splenectomy before reaching adulthood[27]. Importantly, being a fisherman, which was an indicator of past schistosome exposure[28], was positively associated with splenic enlargement. Hb concentrations are already used as a sign of patient stability in emergency care and general screening in routine primary healthcare practice. Hb is a broad indicator that can encompass many risk factors from genetic disorders, infections, poor nutrition, immune disorders, to recent traumatic blood loss, among other causes[29]. Our findings reveal that Hb concentrations are informative for screening for severe schistosomal morbidities. Given its routine measurement in clinical care and low cost, Hb concentration should be investigated in future prospective studies as an early indicator used for triage and risk classification of complex hepatosplenic multimorbidity. As a start, we found differences in WHO anaemia categories[29] related to severe conditions, but additional research is needed to define Hb thresholds to match levels of multimorbidity severity, as tied to progression and prognosis. Once known, national clinical guidelines in countries where schistosomiasis is endemic should consider Hb thresholds in case management protocols for portal hypertension. By contrast, current alcohol use was not associated with any hepatosplenic conditions despite being a regularly assessed risk factor for portal hypertension by clinicians[30].

Easily observable participant characteristics related to demographics, socioeconomic status, and location were also relevant risk factors for hepatosplenic multimorbidity. Males were more likely to have hepatosplenic conditions, in addition to characteristics of being a fisherman and proximity to open freshwater sites. Men are more likely to engage in activities involving water contact, such as fishing, that determine the history of exposure to schistosome infection[28,31], which is of greater relevance than current infection for periportal fibrosis[9,14]. Living in either Buliisa or Pakwach districts had positive effects on overall multimorbidity compared to Mayuge, with the highest likelihood related to Pakwach. This result suggests there is spatial clustering in multimorbidity that needs to be accounted for in future studies. The district effect may also capture high-level risk factors such as access to and the quality of healthcare. Pakwach was the only district in our study that did not have a hospital, instead operating with only one dilapidated and understaffed Health Centre IV, and was often low on emergency medicine kits (saline and other supplies) and blood for transfusion that would be needed to respond to upper gastrointestinal bleeding from gastro-oesophageal variceal bleeding. Pakwach Health Centre IV also lacks essential sonography expertise and ultrasound equipment to non-invasively diagnose gastro-oesophageal varices. The lack of supplies, coupled with inadequate triage and investigation due to a lack of knowledge of risk factors by health workers, may result in premature mortality of patients. The geographical disparities highlighted in this study identify a need for recruitment of additional health staff, education on schistosomiasis-related morbidities, sonography training, and provision of supplies and equipment in the areas with the highest burden of hepatosplenic multimorbidity.

SchistoTrack was designed to include purposely different districts in terms of risk factors and prevalence, so it is unclear whether a district validation shows generalisability or whether it is better to keep a model trained on diverse districts and let it see other settings that are contained in the diversity it is trained on. Splitting data by district influenced predictive performances as expected, and the results suggest mixed trends, but generally, Mayuge was the best predicted district with the highest average AUC-PR and PPV. But, this performance may be due to having most of the cases (the highest condition prevalence) in the other two Western districts, so for the metrics that considered imbalance, there were mostly positive cases to be seen during model training.

We identified risk factors for gastro-oesophageal varices, which indicate severe portal hypertension and pose several challenges for early detection in areas with limited diagnostic resources, including a lack of endoscopy units[32,33]. We focused on the example of gastro-oesophageal varices due to the need for having a concrete clinical endpoint for patient case management in schistosomiasis-endemic areas. High mortality rates of 24% often observed in high-income contexts[34] due to oesophageal variceal bleeding. Even higher mortality rates are assumed in low-income contexts where the burden of gastro-oesophageal bleeding has been poorly documented, and resources for endoscopy, surgical management, beta blockers, and blood transfusion are severely limited. Here, we showed that only older age and lower Hb concentration were positively associated with the likelihood of having gastro-oesophageal varices. Although insignificant, there were non-zero positive effects from HIV, HBV, being a fisherman, and being male, with the largest effect size from HIV. Notably, HIV, being a fisherman, and being male were significantly positively associated with schistosomal periportal fibrosis (Niamey protocol patterns C–F) that precedes portal hypertension complications such as gastro-oesophageal varices. These risk factors have been shown elsewhere to be relevant either due to interacting biological causes (HIV) or representing a history of schistosome exposure (fisherman and gender)[9]. Additional prospective studies are needed to confirm the mechanistic relationship with schistosome-HIV co-infections, as well as to investigate morbidities specific to HIV but not to schistosomiasis.

With regard to overall hepatosplenic multimorbidity or the highest multimorbidity counts, the covariates expected to produce the

highest number of conditions within individuals were low Hb concentration, older age, being male, working as a fisherman, and living in Pakwach district. If a participant has all of these characteristics, then there is a high probability of finding severe enlargements in both the spleen and main portal vein, and a higher number of conditions in general. In our study population, individuals with the most conditions had between six and 13 conditions. The burden of multimorbidity observed here is particularly high, compared to other studies[2] that often find only a few conditions within an individual.

Importantly, we discovered several clinical risk factors from the condition dependencies found in the multitask model for gastro-oesophageal varices. There was a higher probability of an individual having gastro-oesophageal varices when only occluded vessels and fibrosis blocking the main portal vein (Niamey protocol pattern E, patches), or fibrosis at the point of portal bifurcation were observed (Niamey protocol pattern D, ruffing). Community detection on the graph also found gastro-oesophageal varices belonged to the same cluster as all schistosomal periportal fibrosis patterns (Niamey protocol patterns C–F). Interestingly, this cluster appeared clearly separated from a cluster on cirrhosis and its features and complications, as well as another cluster that appeared to entail fatty livers and gall bladder thickening. Future prospective work is needed to assess whether these separate clusters may represent portal hypertension conditions more attributable to schistosomiasis versus HIV or HBV (cirrhosis cluster) versus alcohol use (fatty cluster). There were no associations with current schistosome infection. These findings show that only severe schistosomal periportal fibrosis patterns, which require a long history of schistosome exposure or interactions with co-infections, increase the likelihood of gastro-oesophageal varices. Notably, there were conferred risks from hypersplenism, a severely enlarged main portal vein, and various portosystemic collaterals, including splenic varices and splenorenal shunts. Our study provides a shortlist of conditions and portal hypertension complications that should be prioritised as danger signs for future clinical guidelines seeking to preemptively identify patients at risk of developing gastro-oesophageal varices and manage patients with existing varices.

The graph convolution consisted of dependencies between conditions with extensive evidence of biological realism with respect to expected conditions that should be similar due to biological complications, clinical consequences, or expected pathogenesis/disease progression patterns. A strong connection between Niamey protocol pattern E and F is consistent with the progression of schistosomal periportal fibrosis as staged by the WHO protocol[21]. Spleen enlargement was connected to splenic varices, as splenomegaly typically precedes the development of varices, as congestion in the splenic vein accumulates over time[35]. Splenic varices also have strong associations with gastro-oesophageal varices, reflecting their joint development in response to progressive portal hypertension[36]. The liver surface irregularities were most dependent on cirrhosis, which leads to long-term scarring of liver tissues and commonly presents as a serrated or grossly undulating liver surface[37]. Meanwhile, the infections considered in this study generally do not target any one hepatic lobe; there were multiple connections between the left and right lobes that mostly exhibited matching shrinkage or enlargement. This finding was supported by community detection on the graph, with all enlargements of both liver lobes appearing in one cluster, and all shrunkenness appearing in a separate cluster. Elements of the data acquisition and mode of diagnosis via B-mode ultrasound were also reflected in the graph. For instance, prominent peripheral rings and prominent pipe stems (Niamey protocol patterns C1 & C2) represent different probe cross-sections of the liver and are no different in their interpretation as to disease stage or pathogenesis. Unclear liver fibrosis patterns were connected to fatty livers, as it is commonly known that the excessive brightness of fatty livers can obscure the ability to see other anatomical features within the liver needed for liver fibrosis diagnosis[21].

Conditions unrelated to major infections in the study area (schistosomiasis, malaria, HIV, or HBV) or the majority of the hepatosplenic conditions diagnosed also appear appropriately as isolated nodes; examples include congenital conditions such as situs inversus, describing the reversed positions of organs, and liver cysts. Critically, for the use in real-world applications, the graph convolution was extremely robust to small or large perturbations, suggesting the probabilities of condition dependencies were reliable even if noise existed in the original graph or data (e.g. with false negative or positive diagnoses).

Our study makes several major methodological contributions for modelling multimorbidity. The model presented has richer interpretative elements than standard logistic regression, which cannot provide insight into inter-task strengths, as each condition/outcome is handled independently in the logistic regression. Meanwhile, our model also predicted as well and often better than black-box models that have no interpretability, with only MT-NNs, which has a type of multitask architecture, showing competitive performance. However, MT-NN does not have any clinical interpretability and consists of a larger number of parameters than our model. Meanwhile, logistic regressions struggle with class imbalance and were consistently one of the worst-performing models across the range of metrics. Class imbalance is inherent to hepatosplenic conditions; gastro-oesophageal varices, for example, are an end-stage complication of portal hypertension and often with survivor bias (given associated high mortality rates). This problem was mitigated in multitask settings as each model was able to learn from the models for other, potentially more common conditions. This importance was demonstrated by the relevance of the condition dependencies over covariates for infrequent outcomes such as gastro-oesophageal varices. Multitask learning exposes each prediction to similarly behaving and biologically related conditions to maximise available information. Most conditions, including gastro-oesophageal varices, saw benefits from learning risk factors from other related conditions, as evidenced by high AUC-PRs-to-prevalence ratios, which indicated the proportional improvements in predicting the positive outcomes against a random model. Our study goes beyond the conventional use of graphs for multimorbidity, where condition dependencies are typically inferred only between directly connected nodes[38,39]. The global graph convolution function accounted for edge weights and graph distance, allowing the strength of relationships between nodes further away to also be quantified. Additionally, while previous work has utilised similar variable selection procedures through inclusion probabilities[38,39], they have been treated as discrete binary variables, which are less flexible than continuous inclusion probabilities that could encompass both good and uninformative associations between each covariate and the range of outcomes.

Our study has some limitations. The current framework does not account for task importance and treats all conditions equally, irrespective of severity. Clinically, the inferred dependencies between conditions suggest biological relatedness, but they do not imply causality without further prospective study, and therefore cannot be used to explain liver fibrosis pathogenesis or the origin of portal hypertension being due solely to initial single infections or co-infection interactions over time. The identified risk factors are informative for clinical management, but require additional research to account for genetic risk and identify immunological mechanisms. This study relied mostly on point-of-care ultrasound, which was sufficient for identifying the presence of conditions, but future work may want to consider integration of biomarkers such as platelet counts and liver enzymes to produce fibrosis or clinical severity scores. Additional studies beyond current alcohol use are needed that consider the history of consumption, frequency, type, and quantity. Alcohol use is commonly associated with fatty livers[40], yet no association was found in our study (despite fatty livers appearing in a distinct graph cluster),

which may be a result of the simple binary measure of current alcohol use. Although inferring in a fully Bayesian manner greatly improved the flexibility of the parameters, quantifying associations with heavily skewed posterior distributions, such as those with HIV and HBV, was more difficult. Meanwhile, the graph was utilised to quantify inter-task strengths for finding related conditions as risk factors and allowed us to observe the influence of conditions without including them as input covariates. However, the chosen graph function, despite providing improvements to the prediction, was not as important as the risk factors. Future work should investigate more flexible deployments of the graph to incorporate properties of controlling condition importance and structural learning.

Hepatosplenic multimorbidity is a major global health challenge due to the wide range of biological, social, and spatial causes, high prevalence in resource-constrained settings, and the complexity of identifying condition dependencies to inform clinical guidelines. This study validates the usefulness of Bayesian machine learning in addressing these challenges. We have demonstrated the performance of Bayesian multitask learning for predicting hepatosplenic multimorbidity in schistosomiasis-endemic areas, while the framework has wide generalisability to other multimorbidities. We discovered key risk factors and condition dependencies of 45 schistosomiasis-related hepatosplenic conditions that could be used to develop clinical and WHO guidelines for morbidity identification and case management in endemic countries.

## Methods

### Ethics approvals

Our study complies with all relevant ethical regulations. Data collection and use were reviewed and approved by Oxford Tropical Research Ethics Committee (OxTREC 509-21), Vector Control Division Research Ethics Committee of the Uganda Ministry of Health (VCDREC146), and Uganda National Council for Science and Technology (UNCST HS 1664ES). Written informed consent was obtained for adult participants, with adults consenting on behalf of the children after receiving their informed assent. Ethics approval was obtained from all local ethics committees. Participants were compensated in-kind for their time with a large (1 kg) bar of soap.

### Participant sampling

This cross-sectional study was conducted as a nested design within the SchistoTrack cohort[41] during the first annual follow-up between 17 January and 16 February 2023. Participants were selected from 1952 randomly sampled households from 52 villages across Buliisa, Pakwach, and Mayuge Districts of Uganda; 38 of the villages were sampled in the baseline of 2022[9]. One child aged 5-17 years and one adult aged 18–91 were selected by the household head or spouse and invited for clinical assessments. 3224 individuals were examined with point-of-care ultrasound in 2023. There were 3186 individuals out of 3224 with non-missing ultrasound data, and they were featured in learning the graph[10] utilised for this study. All participants had household demographic covariates data, but 3155 had non-missing data from blood and stool assessments and were considered for analysis in this work. A breakdown of missing data can be found in the participants' flow diagram in Supplementary Fig. S25.

### Hepatosplenic outcomes

The outcomes were hepatosplenic conditions obtained by point-of-care B-mode ultrasound in abdominal view. Philips Lumify C5-2 curved linear array transducers were used with the Philips Lumify Ultrasound Application v3.0 on Lenovo 8505-F tablets with Android 9 Pie. Lossless DICOM images and videos were saved for quality assurance[9]. We assessed a total of 45 hepatosplenic conditions listed in Table 1. Sonographers were blind to participant covariates except for age and gender. The conditions were measured as binary indicators[10],

including focal and diffuse liver fibrosis patterns following the WHO Niamey protocol[21], liver surface irregularities, caudal liver edge assessments, fatty and cirrhotic livers, liver and spleen organometry, portal vein dilation or restriction, portosystemic collaterals, ascites, gall bladder obstruction, among others. For the left and right liver lobes, spleen, and portal vein diameter, we considered abnormality as one or two standard deviations above or below an internal healthy reference population from the same year, split by height, as described in ref. 10. Importantly, all conditions were assigned a severity rating in relation to schistosomal portal hypertension, as shown in the final column. These were determined from expert opinion by a clinical epidemiologist (GFC), sonographer (VA), and gastroenterologist (CKO) as detailed in ref. 10.

### Participant covariates

Participants' covariates were collected through a combination of household surveys and diagnoses by laboratory technicians and nurses. Information was recorded using the Open Data Kit version. 2022.4. These are listed in Table 2, detailed definitions can be found in refs. 9,22. Clinical measurements included *S. mansoni* status as measured by microscopy and eggs per gram, malaria rapid diagnostic test outcomes, HBV rapid diagnostic test outcomes, self-reported HIV, and point-of-care Hb concentration. Although not a model covariate, anaemia status was investigated by constructing indicators based on WHO definitions, where thresholds for mild, moderate, and severe anaemia were considered age, gender, and pregnancy[42]. Socio-demographic information included age, gender, belonging to the majority tribe of their district, belonging to the majority religion of their district, years of educational attainment, and occupations (farmer, fisherman, and fishmonger, with unemployed and other occupations as reference). The home quality score was calculated based on the quality of materials used to build the house. Individuals belonged to a household with social status if any adult member held or previously held a position in the local council, village health team, religious or clan leadership, or influential beach management committees. Additional household variables included number of individuals in the household, number of years household lived in the village, if they owned the home they lived in, number of rooms in the household, current alcohol use based on self-reported consumption within the year preceding recruitment (based on the WHO STEPwise surveys[43]), improved water source defined by if they obtained water from any of protected well or spring, borehole, village tap, and rainwater, number of water activities, and the year of recruitment (2022 or 2023, with the baseline recruitment year of 2022 as reference). Notably, for alcohol use, we also asked about past drinking following the WHO STEPwise approach, but did not consider this variable further, as only 1.43% (45/3155) of all study participants had previously drank and stopped drinking. We did not focus on the frequency or amount of alcohol consumed, as most individuals in our study were not regular drinkers[44]. Finally, locational variables measured the distance (in kilometres) to the nearest freshwater site, government health centre, and the respective districts (Buliisa, Pakwach, with Mayuge as reference). The data was standardised pre-modelling to make priors easier to choose, as detailed in later sections.

### Model specification

We adopted a Bayesian multitask model for predicting the 45 hepatosplenic conditions diagnosed in this study. Multitask models have often been presented as neural networks (e.g. refs. 45–48), but we proposed an architecture and inference framework that was arguably more general and suitable for clinical studies. Importantly, existing work in Bayesian multitask regression[49–51] lacked our level of interpretability. The model was inferred in a Bayesian manner such that the possible relationships between covariates and outcomes were incorporated more

comprehensively as distributions[52], while we were able to maintain interpretations of significance and strength of associations. A Bayesian procedure was implemented to also improve the handling of imbalanced outcomes, as was the case for many severe conditions. Having such uncertainty in the model had been shown to be robust against class imbalance[52], mitigating the need for resampling methods such as SMOTE[53]. Importantly, we designed a new model architecture following the properties of multitasking, introducing ways for parameters to be exposed to multiple outcomes. All factors incorporated led to a final model with far superior predictive performance. Each step of model interpretation and utilisation is summarised in Fig. 1.

Models were developed to predict $M = 45$ possible hepatosplenic conditions within an individual; participant data were indexed by $i = 1, ..., N$. The outcome conditions were represented by indicator vectors $\mathbf{y}_i = (y_{i1}, ..., y_{iM})^\top$, such that $y_{ij} = 1$ if person $i$ has condition $j$, and 0 otherwise. Each participant was represented by $(\mathbf{x}_i, \mathbf{y}_i)$, where we predicted the set of conditions $\mathbf{y}_i$ given the individual covariates $\mathbf{x}_i$. We accounted for dependencies between the conditions through a thresholded multimorbidity graph $\mathcal{G}$ representing positive interrelationships between the 45 hepatosplenic conditions. The graph was learned based on graphical lasso[54], considering the positive edges only with a threshold of 50% validated following the pipeline from a previous study[10].

To progress towards multitask modelling, we started with single outcome logistic regression defined as

$$y_{ij} = g(w_0 + w_1 x_{i1} + \cdots + w_{iP} x_P) + \epsilon_i = g(\mathbf{w}^\top \mathbf{x}_i). \tag{1}$$

In the case when $M$ outcomes ($M > 1$) existed for analysis, we built $M$ models for person $i$ by

$$
\begin{aligned}
y_{i1} &= g(\mathbf{w}_1^\top \mathbf{x}_i) \\
y_{i2} &= g(\mathbf{w}_2^\top \mathbf{x}_i) \\
&\vdots \\
y_{iM} &= g(\mathbf{w}_M^\top \mathbf{x}_i).
\end{aligned}
\tag{2}
$$

For more compact notation, we combined the regression weights/coefficients ($\mathbf{w}_i$) as

$$
\mathbf{y}_i = g\left[ \begin{pmatrix} \leftarrow \mathbf{w}_1^\top \rightarrow \\ \leftarrow \mathbf{w}_2^\top \rightarrow \\ \vdots \\ \leftarrow \mathbf{w}_M^\top \rightarrow \end{pmatrix} \mathbf{x}_i \right]
\tag{3}
$$

$$= g(\mathbf{W}^\top \mathbf{x}_i). \tag{4}$$

Model (4) was the basis of multi-output modelling[55], where it was important to note that $y_{ij} \in \mathbf{y}_i$ continued to be predicted by $\mathbf{w}_j^\top \mathbf{x}_i$ only.

We extended model (4) to a multitask framework by proposing the introduction of two new terms:

$$\mathbf{y}_i = g(f_\alpha(\mathcal{G})\mathbf{W}^\top \operatorname{diag}(\Gamma)\mathbf{x}_i). \tag{5}$$

$f_\alpha$ is a function defined from the graph, and $\Gamma = (\gamma_0, ..., \gamma_P)^\top$ is a vector of probabilities for the $P$ possible covariates. The roles of the two terms were to introduce multitask model structures, as explained in the following sections.

**Multitask graph convolution.** We introduced the graphical function $f_\alpha(\mathcal{G}) \in \mathbb{R}^{M \times M}$ that convolved the models on each outcome. The aim was to replicate the known interrelatedness of biological interactions of hepatosplenic conditions by introducing inherent relational

interdependence between the models of each condition. Without this term, relations between the models would not be incorporated in the model. We chose the graphical function from signal processing[56,57] to act as filtering functions to smooth node data based on the graph

$$f_\alpha(\mathcal{G}) = (\mathbf{I} + \alpha \mathbf{L})^{-1} \tag{6}$$

where $\mathbf{L}$ was the regularised graph Laplacian defined by

$$\mathbf{L} = \mathbf{D}^{-\frac{1}{2}}(\mathbf{D} - \mathbf{A})\mathbf{D}^{-\frac{1}{2}} \tag{7}$$

for adjacency matrix $\mathbf{A}$ and diagonal degree matrix $\mathbf{D}$ based on the graph, and $\alpha$ is an additional positive parameter controlling the influence of the graph.

By multiplying the multi-output model with $f_\alpha(\mathcal{G}) \in \mathbb{R}^{M \times M}$, each outcome would be predicted by not only the corresponding model, but also the models on proximate nodes based on the graph structure. The choice of $f_\alpha$ allowed the construction of continuous weightings that accounted for both edge weights and graph distances. The weights assigned to the nodes were further controlled by the parameter $\alpha$ and optimised as part of the model inference. The $i$th row of $f_\alpha(\mathcal{G})$ provided the weighting across the graph with the $i$th node as centre, ensuring model $i$ is primarily focused on condition $i$ but drawing influence from models on proximate nodes. The size of the weights exhibited a similar scale to probabilities, and were therefore normalised to one so that they could be interpreted as such. Condition $i$ would be predicted by a weighted average of models, with the weighting on proximate nodes indicating the additional likelihood of developing condition $i$ contributed from the other models. The value on node $i$ itself would be interpreted as self-influence, reflecting solely the model developed for that condition and therefore the influence of the covariates on the singular condition. Isolated nodes in the graph represented unrelated conditions and would be treated as independent heterogeneous tasks. The graph filtering function (6) provided weightings for all nodes, incorporating more information than existing studies on multimorbidity networks, which restricted analysis to 1-hop neighbourhoods. As the conditions came in groups based on different diagnoses of sonography (see first column of Table 1 for grouping), some conditions naturally co-occurred; therefore, our analysis also differentiated the two types of influences, with conditions from different groups classified as risk factors, while same group connections were interpreted as co-occurring conditions.

**Multitask covariate learning.** The term $diag(\Gamma) \in \mathbb{R}^{(P+1) \times (P+1)}$ performed the role of multitask covariate learning, with each $\gamma_i$ acting as global inclusion probabilities of the covariates across all models. When combined with the covariates, $diag(\Gamma)\mathbf{x}_i = (\gamma_0, \gamma_1 x_{i1}, ..., \gamma_P x_{iP})^\top$ (assuming the first term in the covariate vector was the intercept). Each $\gamma_i$ was inferred to represent the utility of covariate $i$ for all outcomes as a joint distribution, allowing for covariate information sharing between models. For categorical variables, the categories were encoded as indicators, but we enforced the same inclusion probability across all indicators of the same variable. For instance, if $x_{ic1}$, $x_{ic2}$, and $x_{ic3}$ were possible outcomes of $x_c$, then $diag(\Gamma)\mathbf{x}_i = (\gamma_0, ..., \gamma_c x_{ic1}, \gamma_c x_{ic2}, \gamma_c x_{ic3}, ...)^\top$. Doing this provided a measure of relevance of $x_c$ as a whole instead of each indicator independently.

While the inclusion probabilities acted as global multipliers shared across all models, inferring them in a Bayesian manner expanded their function as a parameterisation of all the possible associations between the covariates against multiple outcomes. Both favourable (high probabilities) and uninformative (low probabilities) associations would be reflected in a single posterior distribution, which was fully incorporated for prediction. The shared nature of $\Gamma$ would also influence the covariates' effect size across tasks; covariates with many

favourable associations would have their effect size increased in other conditions, and vice versa.

The role of inclusion probabilities could also be interpreted in the context of Bayesian variable selection, drawing similarities to the spike-and-slab prior when coupled with the regression coefficients. Each $\gamma_i$ could potentially induce shrinkage effects on the covariates, and as a result, covariates were standardised before model fitting to ensure equal, comparable shrinkage, as consistently practised with common Bayesian variable selection priors[58].

## Prior selection

The multitask model took the following form:

$$\begin{pmatrix} y_1 \\ y_2 \\ \vdots \\ y_M \end{pmatrix} = g\left[ (f_\alpha(\mathcal{G})) \begin{pmatrix} \leftarrow \mathbf{w}_1^\top \rightarrow \\ \leftarrow \mathbf{w}_2^\top \rightarrow \\ \vdots \\ \leftarrow \mathbf{w}_M \rightarrow \end{pmatrix} \begin{pmatrix} \gamma_0 & & & 0 \\ & \gamma_1 & & \\ & & \ddots & \\ 0 & & & \gamma_P \end{pmatrix} \begin{pmatrix} 1 \\ x_1 \\ \vdots \\ x_P \end{pmatrix} \right]. \quad (8)$$

We henceforth refer to $f_\alpha(\mathcal{G})$ as the graph convolution, $\mathbf{W}$ as the regression coefficient matrix, and $\Gamma$ as the inclusion probabilities.

The model was inferred in a Bayesian manner with all parameters given priors, and predictions were found by marginalisation[52,59]. First, inclusion probabilities were assigned priors with support between [0, 1]:

$$\gamma_0, \ldots, \gamma_P \overset{i.i.d.}{\sim} ContinuousBernoulli(\theta = 0.5). \quad (9)$$

The continuous Bernoulli was chosen due to its similarity to the binary counterpart when $\theta$ was close to 0 or 1[60,61], making each $\gamma_i$ easier to interpret as inclusion/exclusion; Beta distributions of various shapes were also tested and led to similar results.

As regression weights $w_{ij}$ could be positive or negative, we used the common Gaussian distribution. From testing, we found the predictive quality was sensitive to the distributional variance, and therefore assigned priors to allow a varied variance

$$w_{ij} \overset{i.i.d.}{\sim} \mathcal{N}(0, \sigma_{ij}^2) \quad (10)$$

$$\sigma_{ij}^2 \overset{i.i.d.}{\sim} \Gamma(2, 2). \quad (11)$$

The positive priors were assigned to the variance of each Gaussian distribution to allow the regression weights to have non-fixed variance to adapt to the uncertainty of the covariates. The Gamma distribution was chosen because of its non-negativeness and flexible shape; its hyperparameters were chosen to have mode 1, made appropriate by the normalisation of the covariates. With a non-fixed variance, we found that Gaussian priors performed similarly to $t$-distributions with a prior-assigned scale parameter (equivalent to the variance of the Gaussian) and could be interchanged.

Concerning the graph convolution with non-negative parameter $\alpha$, we assigned the same positive prior as previously described to provide a suitable starting point during inference

$$\alpha \sim \Gamma(2, 2). \quad (12)$$

Other shapes of Gamma distributions were tested, and we found priors with modal values within the range [1,3] led to similar predictive performances, but larger modes outside of the range led to predictions that were overly smoothed and degraded performance.

## Bayesian inference

Bayesian machine learning is characterised by the inference procedure involving integrating over the uncertainty of all parameters. This was the key advantage of the Bayesian approach, as predictions were based not on single-point estimates of the parameters, but Bayesian averages over all possible parameter configurations weighted by their posteriors. Here, we detail the steps to carry out Bayesian inference on the model.

Model parameters were inferred using Markov Chain (Hamiltonian) Monte Carlo (MCMC)[62,63] using the NUTS kernel from probabilistic programming library Pyro[64]. Let $\Theta = \{\Gamma, \mathbf{W}, \sigma_2, \alpha\}$ represent all parameters of the multitask model $f(\mathbf{x}_i) = g(f_\alpha(\mathcal{G})\mathbf{W}^\top diag(\Gamma)\mathbf{x}_i)$, and data $\mathcal{D} = \{\mathbf{x}_i, \mathbf{y}_i\}_{i=1}^N$. We were interested in obtaining the posterior distribution

$$\mathbb{P}(\Theta | f, \mathcal{D}) \propto \mathbb{P}(\mathcal{D} | \Theta, f) \mathbb{P}(\Theta | f) \quad (13)$$

$$= \underbrace{\mathbb{P}(\mathcal{D} | \Theta, f)}_{\text{Likelihood}} \underbrace{\mathbb{P}(\Gamma) \mathbb{P}(\mathbf{W}) \mathbb{P}(\sigma^2) \mathbb{P}(\alpha)}_{\text{Priors}}. \quad (14)$$

The priors above were defined by Eqs. (9), (10), (11), and (12). For binary classification, the likelihood was the Bernoulli density:

$$\mathbb{P}(\mathcal{D} | \Gamma, \mathbf{W}, \alpha, \sigma^2) = \prod_{i=1}^N \prod_{j=1}^M f(\mathbf{x}_i)_{(j)}^{y_{ij}} (1 - f(\mathbf{x}_i))_{(j)}^{1-y_{ij}} \quad (15)$$

over $N$ participants and $M$ conditions, where $f(\mathbf{x}_i)_{(j)}$ denoted the $j$th element of $f(\mathbf{x}_i) \in \mathbb{R}^M$. We assumed conditional independence of the outcomes in the likelihood, but statistical dependence was induced through the multitask parameters capturing interactions among outcomes. This setup allowed the likelihood to reflect outcome dependencies while retaining a computationally tractable likelihood. This was the optimal choice compared to multivariate logistic[65] or probit[66] likelihoods, which were intractable with the number of outcomes, and the dependence specification was through random effects as covariance structures instead of fixed effects of the multitask parameters.

Predictions on novel inputs $\mathbf{x}_*$ were obtained by computing

$$\mathbb{P}(\mathbf{y}_* | \mathbf{x}_*, f, \mathcal{D}) = \int \mathbb{P}(\mathbf{y}_*, \Theta | \mathbf{x}_*, f, \mathcal{D}) d\Theta \quad (16)$$

$$= \int \mathbb{P}(\mathbf{y}_*, | \Theta, \mathbf{x}_*, f, \mathcal{D}) \mathbb{P}(\Theta | f, \mathcal{D}) d\Theta \quad (17)$$

$$\approx \frac{1}{P} \sum_{i=1}^P \mathbb{P}(\mathbf{y}_*, | \Theta_i, \mathbf{x}_*, f, \mathcal{D}), \Theta_i \sim \mathbb{P}(\Theta | f, \mathcal{D}) \quad (18)$$

$$= \frac{1}{P} \sum_{i=1}^P g(f_{\alpha_i}(\mathcal{G})\mathbf{W}_i^\top diag(\Gamma_i)\mathbf{x}_*), \quad (19)$$

where each set of parameters $\Theta_k = \{\Gamma^k, \mathbf{W}^k, \alpha^k, \sigma^{2^k}\}$ were samples of the posterior obtained by MCMC. Each sample represented a configuration of the model, and they corresponded to the highest posterior density and optimal parameters for model fit. As shown by equation (19), predictions were made by Bayesian averaging $P$ models instead of one single model; this was the signature of Bayesian machine learning that led to far greater flexibility and predictive capacity.

## Covariate significance

Risk factors were obtained from the covariates that exhibited significant associations with the outcome conditions. The likelihood ratio test was used here, as while significance could be determined by examining credible intervals, posterior distributions were skewed and would lead to different conclusions depending on which credible

interval to use. Using credible intervals alone also lacked $p$-values needed for false discovery rate correction, as explained later.

Nested model log-likelihood ratios were well-known to follow the $\chi^2_\nu$ distribution; here, the degree of freedom was dependent on the number of outcomes. The likelihood (15) represented the full model for all outcomes, but for individual conditions, the partial log-likelihoods was computed by fixing $j$ in (15) to the index of the covariate of interest

$$l_j(\mathcal{D}) = \sum_{i=1}^{N} [\, y_{ij} \log(f(\mathbf{x}_i)_{(j)}) + (1 - y_{ij}) \log(f(\mathbf{x}_i)_{(j)})]. \tag{20}$$

Let $l_j(\mathcal{D}_{-k})$ represent the log-likelihood evaluated with covariate $k$ excluded, the test for covariate $k$ against outcome $j$ was computed through the likelihood ratio

$$-2(l_j(\mathcal{D}_{-k}) - l_j(\mathcal{D})) \sim \chi_1^2. \tag{21}$$

More importantly, we were able to measure covariate significance specific to groups of conditions by combining their log-likelihoods. This was useful for examining any multimorbid complications consisting of specific sets of conditions, such as the multiple liver fibrosis patterns indicative of periportal fibrosis (Niamey protocol patterns C–F) or simply clusters obtained from the graph. The likelihood ratio test became

$$\sum_{j \in \Omega} -2(l_j(\mathcal{D}_{-k}) - l_j(\mathcal{D})) \sim \chi_{|\Omega|}^2 \tag{22}$$

where $\Omega$ represented the set of conditions of interest, and the degree of freedom became the size of $\Omega$ because covariate $k$ appeared once in the model on each condition. Summing over all 45 conditions would lead to a measure for overall hepatosplenic multimorbidity.

The large number of tests computed here would lead to a high false discovery rate; thus, we controlled this by converting the $p$-values to $q$-values[67,68] to determine the significance. The categories in occupation and district were treated as separate variables as they consisted of individual coefficients, thereby taking up a degree of freedom each. Thus, there was a total of 28 covariates to test, and we adjusted for the $28 \times 45 = 1260$ tests for all individual conditions, and 28 tests for grouped multimorbidity.

## Risk factor association

We interpreted the strength and direction of risk factor association by converting model posteriors to odds ratios. Each element of the model - graph convolution, regression coefficients, and inclusion probabilities - had individual posteriors, but we used the joint distribution of the three combined to reflect the scale of the data. Odds ratios were outcome-specific, such that for covariate $j$, we take the entry $[f_\alpha(\mathcal{G})\mathbf{W}^\top diag(\Gamma)]_{ij}$ to find the association with condition $i$. Division by the standard deviation of each covariate was applied such that the odds could be interpreted in the original units (undoing the data standardisation prior to model inference).

More precisely, let $m_{ij}$ represent the $ij$-th entry of $[f_\alpha(\mathcal{G})\mathbf{W}^\top diag(\Gamma)]$, and $\{m_{ij}^k\}_{k=1}^P$ represent the MCMC samples, we examined the empirical distribution of the odds ratio by the points $\{\exp(m_{ij}^k/\sigma_j)\}_{k=1}^P$ where $\sigma_j$ was the standard deviation of covariate $j$. When considering grouped conditions, we combined the MCMC samples for each condition to provide the overall effect.

The posterior densities were generally skewed and asymmetric, thus, we reported the median when inspecting individual covariates, but used the mean for overall effects. Credible intervals to 95% were computed using the highest posterior density[69], which were better equipped to handle skewed distributions than equal-tailed intervals.

## Graph convolution under perturbation

The graph convolutional matrix was tested under perturbation of the graph. We considered two scenarios, first involving random addition and removal of edges, and adding random edges only. We tested proportions of edges randomly perturbed at 5%, 10%, and 20%, which led to a balance between noise and informative variation on the weights of the convolution. For edge addition, pairs of nodes were randomly selected and given edge weights that were randomly bootstrap-sampled from the list of existing edge weights. Random perturbation was run 100 times to produce the overall effect on the edges. We examined this mainly in the context of dependencies of gastro-oesophageal varices.

## Model prediction and evaluation

Prior analysis had been performed using 100% of the data, but to compare the predictive performance of multitask modelling against other frameworks, we now divided participants into training and testing sets of sizes 50% each (1593). The sets were randomly sampled, and we completed this ten times to obtain an overall evaluation of the predictions. This procedure was chosen instead of cross-validation because some conditions occurred infrequently, and for these cases, random sampling would lead to more splits with at least one positive case appearing in the test set, allowing the model to be evaluated on infrequent conditions more often.

We ran various versions of our multitask modelling, where, in addition to the main model introduced, we also tested multitask using only the graph convolution corresponding to model (8) without the covariate learning term, and multitask using only covariate learning corresponding to model (8) without the graph convolution matrix. The main model in this study featured both multitask learning elements (8). Additionally, a number of machine learning models were tested as baselines: logistic regression using AIC variable selection on each outcome independently, XGboost for each outcome separately, multi-output random forest that modelled all outcomes jointly, and single and multitask shallow neural networks (as depicted in ref. 18), where the single task trained separate neural networks for each outcome while the multitask trained one neural network to predict 45 outcomes jointly.

For our Bayesian models, the training stage involved sampling from the posterior distributions $\mathbb{P}(\Theta|\mathcal{D}, f)$ using (Hamiltonian) MCMC, providing us with a set of posterior sample points for all parameters. For each split, we ran 500 warm-up steps followed by 1000 posterior sampling steps. To make predictions given the test participants, the MCMC samples were used to compute the Monte Carlo estimator (19).

The model outputs were probabilities, so evaluations were done firstly using area under the receiver operating curve (AUC), and AUC-precision-recall (AUC-PR), as well as binary-based metrics F1 (binary and macro), precision, recall, positive and negative predicted values (PPV and NPV), and balanced accuracy after conversion. For binary-based metrics, we considered choosing the thresholds on the predicted probabilities using two different clinical measures: 1. the threshold leading to the highest F1 score was used to compute the precision, recall, and balanced accuracy (both the binary and macro versions in sklearn were evaluated), 2. the threshold leading to 90% sensitivity was used to evaluate PPV and NPV.

## Reporting summary

Further information on research design is available in the Nature Portfolio Reporting Summary linked to this article.

# Data availability

The raw data are protected and are not available due to data privacy laws. The metadata generated in this study are provided in the Supplementary Information.

## Code availability

Synthetically generated data is provided based on random sampling of the covariates and conditions to allow the running of the code. The model implementation code is shared as supplementary material and can be found in Supplementary Code 1.

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

## Acknowledgements

We are thankful for the involvement of our study participants and the SchistoTrack teams, especially the surveyors, nurses, sonographers, and laboratory technicians. We also like to thank the Uganda Ministry of Health, local district leaders, focal health workers, and village health teams. Special thanks also to the Oxford team for the fieldwork, data wrangling, everyday discussions, and feedback. This research was funded in whole, or in part, by the UKRI EPSRC [EP/X021793/1]. For the purpose of Open Access, the author has applied a CC BY public copyright licence to any Author Accepted Manuscript version arising from this submission. NDPH Pump Priming Fund, John Fell Fund, Robertson Foundation, UKRI EPSRC (EP/X021793/1) grants were awarded to G.F.C.

## Author contributions

Conceptualisation: G.F.C. and Y.C.Z. Data curation: Y.C.Z., V.A., J.B.O., B.N., C.K.O., N.B.K. and G.F.C. Formal analysis: Y.C.Z. Investigation, methodology, visualisation: Y.C.Z. Writing - original draft: Y.C.Z. and G.F.C. Validation: Y.C.Z. and G.F.C. Writing - review and editing: Y.C.Z., V.A., J.B.O., B.N., C.K.O., N.B.K. and G.F.C. Funding acquisition and supervision: G.F.C. Resources: G.F.C.

## Competing interests

The authors declare no competing interests.
