## [Peer Review file · Nature Communications]

Bayesian machine learning enables discovery of risk factors for hepatosplenic multimorbidity related to schistosomiasis

Corresponding Author: Professor Goylette Chami

Version 0:

Reviewer comments:

Reviewer #1

(Remarks to the Author)

Zhi and coworkers present a manuscript with an impressive and comprehensive approach to assess hepatosplenic multimorbidity in an Ugandan population, including ultrasound-data.

Overall, the manuscript is written very well, the analysis was conducted reliably and is offering new insights.

The manuscript should be encouraged to be accepted after several revisions:

A major issue seems to be alcohol-induced hepatosplenic multimorbidity as outlined below

Issues:

-hepatosplenic multimorbidity should be explained in further detail. It is recommended, for the readers' comfort, to include in the introduction the spectrum of diseases (that are prevalent in the patients from this region) included in this definition (e.g. liver cirrhosis and hepatosplenic multimorbidity)

-How was liver fibrosis diagnosed/defined (line 103)

-Line 308, „burst gastro-oesophageal varices“. More commonly the term oesophageal variceal bleeding is used.

-Are any markers and scores for cirrhosis (e.g. FIB4-Score) and portal hypertension (e.g. Platelet-count) available? If possible, they should be included in the analysis.

-The outcome and data on the specific disease entities in hepatosplenic multimorbidity should be analysed on each entity (Schistosomiasis, hepatitis B, liver cirrhosis etc.) in more detail and if possible, analysed separately.

Quite surprising! Current alcohol use was not associated with any hepatosplenic conditions

Even though, it seems the only information on alcohol consumption was current use yes or no (respectively „based on self-reported consumption within the year preceding recruitment“)

How was current alcohol use defined (How many drinks per day over which time period? Is data available on past alcohol consumption? Alcohol consumption yes or no does not seem to be sufficient enough.

How prevalent is alcohol abuse in this population? Could further data be provided? Could further alcohol-specific analyses of this dataset provide further insight?

More information is needed on this matter.

This might, if no further information is available, be a strong limitation of this analysis (as it is already included in line 291/292) and should be explained in detail, how this confounder is not severely limiting the conclusions drawn from the analysis.

-Are we sure what endpoints that are clinically relevant were assessed (hepatosplenic multimorbidity is a term not really helpful for a clinician taking care for the patients, as it contains quite different disease entities)?

This should be further discussed in the manuscript

Can it be distinguished in the analysis which patients had hepatosplenic multimorbidity caused by cirrhosis (e.g. due to alcohol abuse or hepatitis B) and which portal hypertension due to schistosomiasis, as well as other causes? More comments on this matter should be included.

(Remarks on code availability)

Reviewer #2

(Remarks to the Author)

Summary

This manuscript presents an innovative Bayesian multitask learning framework for modeling 45 hepatosplenic conditions in a large schistosomiasis cohort. The integration of graph-informed condition dependencies, global covariate inclusion probabilities, and fully Bayesian inference is technically sophisticated and well motivated. The dataset is unique and clinically valuable, and the approach yields meaningful biological and epidemiological insights.

The work has strong potential impact, but several aspects must be strengthened for publication. In particular, while the authors emphasize interpretability, which is appropriate for the clinical domain, the methodological evaluation needs improvement. Broader comparative analyses and more imbalance-aware performance metrics would make the contribution significantly more compelling.

Major Comments

1. Performance comparison should go beyond interpretability-focused baselines

The authors explicitly position their method as an interpretable alternative to opaque black-box models. This is a valid and important goal. However, the paper would be considerably more impactful if it also demonstrated where the proposed model stands relative to commonly used multitask or multilabel learning approaches.

Even if the authors prefer simpler or more interpretable models, showing how their Bayesian architecture compares to at least a subset of widely used baselines, such as multi-output Random Forests, XGBoost, classifier chains, or a shallow shared-bottom neural network, would provide crucial context. These comparisons need not detract from the interpretability focus; rather, they would strengthen the overall contribution by showing either (i) competitive performance with substantially greater interpretability, or (ii) superior performance in the setting of rare outcomes.

This addition would broaden the paper's relevance and demonstrate the practical advantages of the proposed Bayesian framework beyond conceptual elegance.

2. AUC alone is not appropriate given the extreme class imbalance

Many of the modeled conditions occur in less than 5% of individuals, and several in less than 1%. In these cases, ROC-AUC is known to be an insufficient metric, as it remains high even when precision for rare events is very low. Since the authors argue that their Bayesian model handles imbalance more effectively, this claim should be supported with metrics designed for imbalanced classification.

The manuscript would benefit from reporting precision–recall curves, PR-AUC (or average precision), precision, recall, F1-score, and balanced accuracy. These metrics are standard in modern evaluation of imbalanced data and are essential for understanding model behavior in clinically relevant scenarios.

3. Threshold-based evaluation is needed for clinical relevance

Because the authors position the model as a clinical decision-support tool, threshold-based evaluation is essential. Clinicians need to know how reliable a prediction is when the model actually issues it. Metrics such as Positive Predictive Value (PPV) (when the model predicts a patient is positive, how likely is it that the patient truly has the condition?) and Negative Predictive Value (NPV) (when the model predicts a patient is negative, how likely is it that the patient truly does not have the condition?) are crucial. These should be reported at clinically meaningful operating points, such as sensitivities fixed at 90% or 95%. This information determines whether the model's improvements translate into practical utility, especially for rare conditions where AUC can be misleading despite poor PPV.

To make this concrete, consider a condition with <1% prevalence. A model may achieve an AUC of 0.90 yet still have a PPV of only a few percent at clinically practical thresholds, meaning that the vast majority of "positive" predictions would be false alarms. Without PPV, NPV, the false-positive burden at fixed sensitivities, and appropriate precision–recall evaluation, it is impossible to determine whether the reported performance improvements or the conditional relationships inferred by the model translate into meaningful diagnostic value.

4. Graph-derived conditional dependencies must be explicitly validated and interpreted

While the Discussion briefly relates some inferred condition relationships to known hepatosplenic pathways, this evaluation remains qualitative. Given that the graph-convolution component is central to the model's interpretability claims, the manuscript should more explicitly and systematically analyze whether the conditional relationships inferred by the model align with established clinical mechanisms, such as known sequences of fibrosis progression, portal hypertension development, and splenic involvement. It would be valuable for the authors to indicate where the learned dependencies support existing pathophysiological understanding and where they diverge. Without such analysis, it is difficult to determine whether the smoothing imposed by the graph enhances biological realism or primarily reflects statistical correlation structure.

5. Evidence for generalizability would strengthen the contribution

While external validation datasets for schistosomiasis are scarce, a geographically stratified validation (training on two districts and testing on the third) would provide valuable insight into model transportability. If this is not feasible, some robustness analysis (bootstrapping, district-weighted resampling, or sensitivity to graph structure) would still enhance the reliability of the findings.

6. Data availability and reproducibility

While I understand the ethical and privacy constraints around participant-level data, it would be helpful for the authors to clarify whether any form of anonymized, de-identified, or synthetic dataset could be shared to facilitate reproducibility. If full individual-level data cannot be released due to ethics board restrictions or data governance rules, the authors might still consider providing a minimally de-identified feature matrix (removing geographic, demographic, and other potentially identifying variables) or a synthetic dataset generated to match the statistical properties of the original cohort. Either option would enable readers to reproduce the modeling pipeline and evaluate the proposed method more effectively.

Minor Comment

Figure 3, including its schematic and legend, should be more explicative. This figure introduces important components of the model but currently requires substantial inference from the reader. Additional labels or a clearer narrative description would improve clarity.

Conclusion and Recommendation

This manuscript addresses a highly relevant public health problem with an original, well-designed Bayesian modeling framework. The approach is conceptually strong, and the epidemiological insights are of genuine interest. Incorporating broader performance baselines, imbalance-aware metrics, clinically grounded threshold analysis, and explicit interpretation of graph-derived dependencies will significantly strengthen the manuscript.

I therefore recommend Major Revision.

(Remarks on code availability)

I examined the shared materials (README.md and pyro_mcmc.py). Although the core Bayesian multitask model is implemented, the current repository does not allow readers to reproduce the results reported in the manuscript. The main script depends on a non-included module (data_loader) and on participant-level data that are not provided, so the pipeline cannot be executed end to end. The example at the bottom of the script does not run successfully either. It attempts to compute AUROC values but references a variable (test_id) that is undefined when no train-test split is created, meaning that the example cannot produce the metrics shown in the paper.

The underlying probabilistic model appears correctly implemented, but without a working example, without any form of anonymized or synthetic dataset, and without an evaluation pipeline that mirrors the manuscript, the results are not currently reproducible. To meet the reproducibility expectations of the journal, the authors should provide a functional example using either anonymized or synthetic data, ensure that AUROC values and other metrics can actually be computed and saved, and expand the evaluation to include imbalance-aware metrics. A more complete README and more robust covariate handling would also improve clarity and usability.

Reviewer #3

(Remarks to the Author)

This study deploys an interpretable Bayesian multitask learning framework to jointly model 45 hepatosplenic ultrasound outcomes in a rural Ugandan cohort. By combining clinical tests + sociodemographic + access/location covariates with a graph-convolution over an outcome-dependency network and continuous inclusion probabilities for covariates, the authors both (i) improve discrimination—especially for rare, severe outcomes—and (ii) surface clinically plausible risk patterns (older age↑; low Hb↑; female sex protective; higher risk among fishers). The work is methodologically thoughtful and practically relevant to triage in resource-limited settings. So it is suggested to make some revisions with following comments:

1. Primary estimation & interpretability.

Most substantive findings in the text derive from per-outcome logistic regressions (ORs), while the proposed multitask model is judged mainly by AUC. If interpretability is central, the authors should report posterior log-odds (converted to ORs) with

95% HPD intervals for key covariates within the multitask model (e.g., $f_{\alpha}(G) W^{\top} \text{"diag"} (\Gamma)$ mapped to ORs after reversing standardization). Please also explain how Γ (continuous inclusion) relates to effect sizes and whether high- Γ variables consistently exhibit larger posterior odds shifts across tasks.

2. Validation design (80/20 and beyond).

Validation design should also be expanded and clearly described. If the data are split 80/20, please report repeated stratified 80/20 evaluation over many resamples; additionally, include a geographical transport test (leave-one-district-out) and, if feasible, a temporal validation to probe stability over time. For rare outcomes, present PR-AUC, F1-macro, and confusion matrices at clinically meaningful thresholds. Because clinical deployment hinges on threshold choices, decision-curve analysis would add practical value by quantifying net benefit across plausible risk thresholds.

3. Multimorbidity patterns (go beyond “better AUC”).

Since multimorbidity modeling is a stated novelty, I encourage the authors to move beyond “better AUC” and explicitly characterize comorbidity patterns induced by the model. Identify covariate scenarios that maximize predicted multimorbidity counts (e.g., low Hb combined with older age and longer distance to facilities); present patient-level archetypes by clustering predicted outcome vectors; and summarize which outcomes “donate” the most information to others via the graph by reporting row-wise weights from the graph convolution. Interpreting these pathways in light of hepatosplenic pathophysiology would substantiate the claim that the model surfaces clinically actionable “danger signals.”

Finally, robustness and reporting can be strengthened. Please conduct graph sensitivity analyses by perturbing edge weights/thresholds or re-learning the graph and assessing stability of performance and danger-signal rankings; perform prior sensitivity for the graph-strength and variance components; specify missing-data handling; and document prevalence and any class-imbalance strategies.

(Remarks on code availability)

Version 1:

Reviewer comments:

Reviewer #1

(Remarks to the Author)

After the corrections, the manuscript can now be recommended for publication

(Remarks on code availability)

Reviewer #2

(Remarks to the Author)

I thank the authors for their careful and thorough responses to the reviewers' comments. The revisions have satisfactorily addressed the previously raised concerns, and the manuscript has been substantially improved in clarity, methodological transparency, and interpretation of the results.

In its current form, the study is sound, well presented, and makes a meaningful contribution to the field. I have no further comments and recommend the manuscript for publication.

Minor comment:

In Tables 4 and S7 (Model performance comparison), the columns appear to lack sufficient spacing or margins, which makes some values difficult to read. I recommend adjusting the table formatting to improve readability.

(Remarks on code availability)

The authors have made the code publicly available and runnable, with adequate documentation. Due to privacy and governance constraints associated with the clinical data, the original dataset cannot be shared. To address this, the authors provide a synthetic dataset that allows users to run the pipeline and verify code functionality. While the published results are therefore not fully reproducible using the shared data, the implementation and workflow are transparent and reproducible, and the code constitutes a usable resource for the community.

Reviewer #3

(Remarks to the Author)

The revisions have significantly enhanced the clarity and rigor of the analysis. The authors have provided more detailed explanations, particularly regarding the definitions and diagnostic methods for hepatosplenic multimorbidity, which is greatly appreciated. The clarification on alcohol consumption, the inclusion of additional validation steps, and the expanded discussions on model performance metrics further strengthen the manuscript.

I am satisfied with the revisions made and agree that the manuscript is now much clearer and better aligned with clinical

relevance. All of the points raised in my previous review have been adequately addressed.

(Remarks on code availability)

Yes code provided can be run, in which the main implementation was instored in pryomcmc_synth.py file. The code provide a README file,with enough instructions , including: node and edge lists as csv, for constructing the multimorbidity graph required for the model, and synthetic condition outcomes and covariates csv, to allow the code to run.

Reviewer 1

Comment:

Zhi and coworkers present a manuscript with an impressive and comprehensive approach to assess hepatosplenic multimorbidity in an Ugandan population, including ultrasound-data.

Overall, the manuscript is written very well, the analysis was conducted reliably and is offering new insights.

The manuscript should be encouraged to be accepted after several revisions: A major issue seems to be alcohol-induced hepatosplenic multimorbidity as outlined below

Response We appreciate the feedback and recognition of the comprehensive nature of our approach. We have incorporated all suggestions in our revisions, which have improved the clarity from a clinical perspective of our manuscript.

Comment:

hepatosplenic multimorbidity should be explained in further detail. It is recommended, for the readers' comfort, to include in the introduction the spectrum of diseases (that are prevalent in the patients from this region) included in this definition (e.g. liver cirrhosis and hepatosplenic multimorbidity)

Response: Thank you for pointing out that a more direct definition would have been helpful in the introduction. We have added to the second paragraph the following (with citations in text) "The most common hepatosplenic conditions that occur in schistosomiasis-endemic populations include periportal fibrosis, early cirrhosis-like or hepatitis-like livers, splenomegaly, moderately shrunken or enlarged livers, and thickened gall bladder walls—all of which are often accompanied by clinical symptoms such as diarrhea, abdominal pain, anaemia, jaundice, and melena. Hepatosplenic multimorbidity due to chronic schistosome infections encompasses two or more interacting or co-occurring conditions ranging from combinations of less severe forms of periportal fibrosis and changes in spleen or liver organometry to inclusion of life-threatening complications of gastro-oesophageal varices and liver cirrhosis."

Comment:

How was liver fibrosis diagnosed/defined (line 103)

Response: In this particular sentence, liver fibrosis was defined as periportal fibrosis and specifically Niamey protocol pattern C for the diagnosis. We have clarified in text and ensured the Niamey protocol is cited in the Methods. We note in the last paragraph of the introduction (as well as in the methods) that diagnosis was by B-mode point-of-care ultrasound. To adhere to journal guidelines, we keep additional details to the Methods that comes after the results.

Comment:

Line 308 "burst gastro-oesophageal varices". More commonly the term oesophageal variceal bleeding is used.

Response: Corrected.

Comment:

Are any markers and scores for cirrhosis (e.g. FIB4-Score) and portal hypertension (e.g. Platelet-count) available? If possible, they should be included in the analysis.

Response: Unfortunately, as this study relied on point-of-care ultrasound, we did not have any biomarkers available for analysis. We now mention this single diagnostic as a limitation in the Discussion and that the inclusion of biomarkers such as platelets and liver enzymes (e.g. AST or ALT) may better enable severity grading.

Comment:

The outcome and data on the specific disease entities in hepatosplenic multimorbidity should be analysed on each entity (Schistosomiasis, hepatitis B, liver cirrhosis etc.) in more detail and if possible, analysed separately.

Response: Thank you for this comment. We provide models on each condition entity and their risk factors for any condition that has at least one significant risk factor in Table 3, we also have the breakdown for the individual condition risk factors separated by adults and children in Tables S3 and S4 respectively. There is a detailed presentation of these results in the second paragraph of the first Results section. We also now add results to indicate what combination of risk factors (e.g. infection, demographics, etc) produce the highest multimorbidity counts to give more detail on the multimorbidity.

Comment:

Quite surprisingly current alcohol use was not associated with any hepatosplenic conditions. Even though, it seems the only information on alcohol consumption was current use yes or no (respectively "based on self-reported consumption within the year preceding recruitment") How was current alcohol use defined (How many drinks per day over which time period? Is data available on past alcohol consumption? Alcohol consumption yes or no does not seem to be sufficient enough.

Response: Thank you. We agree that this variable requires more detail. We have noted now in the Methods that we used the internationally validated survey tool from the WHO STEPwise Approach. We also add the following information. "Notably, for alcohol use, we also asked about past drinking following the WHO STEPwise approach, but did not consider this variable further as only 1.3% (40/3155) of all study participants had previously drunk and stopped drinking." Moreover, we note that we only focus on current drinking without respect to amount as most individuals in our study population were only occasional drinkers from our detailed analysis elsewhere (citation provided in text). Moreover, alcohol use is most often associated with fatty livers, so it is for specific outcomes that we might expect the effects to be underestimated. We note this as a limitation now in the Discussion and that future work also is needed to better define these variables to get cultural information on alcohol use, frequency, and amounts of alcohol consumed even if only occasionally.

Comment:

How prevalent is alcohol abuse in this population? Could further data be provided? Could further alcohol-specific analyses of this dataset provide

further insight? More information is need on this matter.

This might, if no further information is available be a strong limitation of this analysis (as it is already included in line 291/292) and should be explained in detail, how this confounder is not severely limiting the conclusions drawn from the analysis.

Response: We agree that this is an important area for future research, and as such have initiated a detailed investigation in the next annual timepoint for SchistoTrack to collect information on community consumption patterns, alcohol purchase behaviours, local definitions of alcohol use (both identified socially through the community and by clinicians handling these patients), and more information on type, quantity, and frequency. This information, though not at the individual level, would provide high level information to explain the alcohol use patterns.

Comment:

Are we sure what endpoints that are clinically relevant were assessed (hepatosplenic multimorbidity is a term not really helpful for a clinician taking care for the patients, as it contains quite different disease entities)? This should be further discussed in the manuscript

Response: We agree that the clinical endpoint of focus is important for patient management. In this respect, we provide a breakdown of individual conditions and we focus throughout the manuscript on the example of gastro-oesophageal varices and its concurrent/interacting conditions. We add this further justification to the discussion paragraph we discuss the results of gastro-oesophageal varices. "We focused on this example due to the need of having a concrete clinical endpoint for patient case management in schistosomiasis-endemic areas. High mortality rates of 24% often are observed across low and high income contexts due to oesophageal variceal bleeding. Even higher mortality rates are assumed in low-income contexts where the burden of gastro-oesophageal bleeding has been poorly documented and resources for endoscopy, surgical management, beta blockers, and blood transfusion are severely limited."

Comment:

Can it be distinguished in the analysis which patients had hepatosplenic multimorbidity caused by cirrhosis (e.g. due to alcohol abuse or hepatitis B) and which portal hypertension due to schistosomiasis, as well as other causes? More comments on this matter should be included.

Response: Thank you for this suggestion, which initiated additional exploration. We looked at grouping the conditions on the graph using traditional methods of clustering (community detection) and found that there were three distinct groups/clusters. We have added results to the Graph convolution section as follows. "To give an alternative view on condition clusters using the graph convolution network, we show the grouping of conditions found through community detection on the graph (Fig. S24). Three main clusters were found. One cluster mainly contained measurements of liver enlargement, fatty livers, and other abnormalities not specifically attributable to infections and potentially due to alcohol use. Another cluster that included cirrhosis had conditions that represent features, complications, of consequences of cirrhosis that may be attributable to alcohol use, HIV, or HBV such as rounded liver edges, shrunken liver lobes, and gross

liver surface irregularities. The third cluster encompassed mostly schistosomal periportal fibrosis Niamey patterns, its features such as changes in the main portal vein, and portal hypertension consequences including gastro-oesophageal varices and splenorenal shunts."

We also added the following to the discussion. "Community detection on the graph also found gastro-oesophageal varices belonged to the same cluster as all schistosomal periportal fibrosis patterns (Niamey protocol patterns C - F). Interestingly, this cluster appeared clearly separated from a cluster on cirrhosis and its features and complications as well as another cluster that appeared to entail fatty livers and gall bladder thickening. Future prospective work is needed to assess whether these separate clusters may represent portal hypertension conditions more attributable to schistosomiasis versus HIV or HBV (cirrhosis cluster) vs alcohol use (fatty cluster)."

Though, we do not want to over-interpret these clusters as we think despite the distinction being ideal, it is best left to prospective analyses where issues of temporality and sequence of exposures can be dealt with to avoid reverse causality. For the portal hypertension that may be due more specifically to schistosomiasis, we do try to make an argument in the Discussion showing how periportal fibrosis patterns (focusing on schistosomiasis-specific liver fibrosis patterns) are the most important condition risk factors for gastro-oesophageal varices (a clinical indication of portal hypertension). But, overall, we hypothesize (as has been shown elsewhere in Anjorin et al. 2024 *Lancet Microbe*) that we think the cirrhosis and portal hypertension actually may arise due to interactions between hepatitis b, HIV, and schistosomiasis. Again, these interactions would need to be untangled in prospective work. We mention this in a few places in the Discussion but now make it clear in the limitations. "Clinically, the inferred dependencies between conditions suggest biological relatedness, but they do not imply causality without further prospective study, and therefore cannot be used to explain liver fibrosis pathogenesis or the origin of portal hypertension being due solely to initial single infections or co-infection interactions over time."

Reviewer 2

Comment

This manuscript presents an innovative Bayesian multitask learning framework for modeling 45 hepatosplenic conditions in a large schistosomiasis cohort. The integration of graph-informed condition dependencies, global covariate inclusion probabilities, and fully Bayesian inference is technically sophisticated and well motivated. The dataset is unique and clinically valuable, and the approach yields meaningful biological and epidemiological insights.

The work has strong potential impact, but several aspects must be strengthened for publication. In particular, while the authors emphasize interpretability, which is appropriate for the clinical domain, the methodological evaluation needs improvement. Broader comparative analyses and more imbalance-aware performance metrics would make the contribution significantly more compelling.

Response: We really appreciate the comments on the innovative aspects of the framework, and clinical value of the data and interpretive analysis. We are also thankful that the reviewer drew our attention to imbalance-aware metrics that will greatly improve the comparative analysis of the paper. All suggestions have been incorporated and have strengthened the manuscript.

Comment:

1. Performance comparison should go beyond interpretability-focused baselines

The authors explicitly position their method as an interpretable alternative to opaque black-box models. This is a valid and important goal. However, the paper would be considerably more impactful if it also demonstrated where the proposed model stands relative to commonly used multitask or multilabel learning approaches.

Even if the authors prefer simpler or more interpretable models, showing how their Bayesian architecture compares to at least a subset of widely used baselines, such as multi-output Random Forests, XGBoost, classifier chains, or a shallow shared-bottom neural network, would provide crucial context. These comparisons need not detract from the interpretability focus; rather, they would strengthen the overall contribution by showing either (i) competitive performance with substantially greater interpretability, or (ii) superior performance in the setting of rare outcomes.

This addition would broaden the paper's relevance and demonstrate the practical advantages of the proposed Bayesian framework beyond conceptual elegance.

Response: We are thankful for the reviewer for emphasizing the broader context of performance comparison. We have added a number of model comparisons, which have further strengthened our results to show that our proposed model has very strong predictive capabilities relative to other models. We introduced XGBoost and multi-output

random forest, and shallow deep learning based models of multi and single task learning as described in Caruana [18]; standard logistic regression remains another core comparison for our model given its standard use in epidemiology. We compared the models by inspecting the newly added Table 4 in the manuscript, which contains a number of relevant metrics accounting for class imbalance and relevant for clinical interpretation; some additional less informative metrics can be found in Table S7. These tables replaced the original Fig. S6 in the main paper and Results section "Multimorbidity prediction" has been updated with results as described below.

The performance of our Bayesian multitask model and its variants still remained amongst the best, with only the multitask neural network being competitive based on a number of the metrics including AUC, AUC-PR, binary F1 and precision, and PPV. On other metrics such as Macro F1 and negative predicted value (NPV), there were less noticeable differences as all models performed similarly well. It is worth highlighting the competitive performance from the multitask neural network, which also supports our claim of the advantage of joint modelling. On individual conditions, we added model comparisons on AUC-PR and PPV in addition to AUC. Fig. 3, which previously compared AUCs of individual conditions, has been changed to AUC-PRs-to-prevalence ratio and compared to the prevalence of the conditions. As there were many models, comparisons between models for individual conditions are left to Supplementary Results. To focus the reader's attention, we chose to compare our main model against logistic regression (most interpretable) and multitask neural networks (best performing baseline) in the main text—evaluated using AUC (Fig. S7 - S8), AUC-PR (Fig. S9 - S10), and PPV (Fig. S11 - S12). We agree that AUC can be less informative on imbalanced data and are now mainly examining AUC-PR and PPV. Our model had higher AUC-PRs on 82.22% (37/45, three significant) of conditions than logistic regression, and 73.33% (33/45, one significant) of conditions against multitask neural network. Meanwhile, the PPVs of our model was higher than logistic regression on 84.44% (38/45, four significant) of conditions, and higher on 66.67% (30/45, one significant) of conditions compared to multitask neural network. These results indicate that our model was better at predicting each individual condition by a small margin, rather than being significantly better on a select few. Even though the overall performance is very similar on imbalance aware metrics, logistic regression and any neural network cannot provide the rich interpretable elements of our model. No baseline models can comment on condition dependence, while neural networks and other black-box models cannot identify risk factors, and strength and direction of association.

Comment:

2. AUC alone is not appropriate given the extreme class imbalance

Many of the modeled conditions occur in less than 5% of individuals, and several in less than 1%. In these cases, ROC-AUC is known to be an insufficient metric, as it remains high even when precision for rare events is very low. Since the authors argue that their Bayesian model handles imbalance more effectively, this claim should be supported with metrics designed for imbalanced classification.

The manuscript would benefit from reporting precision–recall curves, PR-AUC (or average precision), precision, recall, F1-score, and balanced accuracy. These metrics are standard in modern evaluation of imbalanced data

and are essential for understanding model behavior in clinically relevant scenarios.

Response: We thank the reviewer for pointing this out as we agree that AUC alone does not uncover the full behaviour of our models. As noted in the response to the previous comment, we have added a number of model comparisons. Along with those models, we added metrics of AUC-PR, F1 (binary and macro), precision, recall, PPV, NPV, and balanced accuracy. For reference on the reported AUC-PR, the prevalence of all outcomes is 0.046. We found the AUC-PR achieved by our model to be a good level for such low prevalences, as this has been observed previously, such as the example in [1] (reference only in response document) where the authors worked with a disease of 4% prevalence and off-the-shelf models achieved similar AUC-PRs. Thus, we have chosen to base our main comparisons using AUC-PR instead of AUC. For threshold based metrics, a threshold was chosen based on the what has been considered clinically meaningful: precision and recall were computed based on the threshold that led to the highest F1 score for binary and macro respectively; F1 was chosen here because it provides a balanced measure of both metrics. PPV and NPV are determined using a second threshold selection method based on 90% sensitivity, which is further detailed in the next response.

Comment:

3. Threshold-based evaluation is needed for clinical relevance

Because the authors position the model as a clinical decision-support tool, threshold-based evaluation is essential. Clinicians need to know how reliable a prediction is when the model actually issues it. Metrics such as Positive Predictive Value (PPV) (when the model predicts a patient is positive, how likely is it that the patient truly has the condition?) and Negative Predictive Value (NPV) (when the model predicts a patient is negative, how likely is it that the patient truly does not have the condition?) are crucial. These should be reported at clinically meaningful operating points, such as sensitivities fixed at 90% or 95%. This information determines whether the model’s improvements translate into practical utility, especially for rare conditions where AUC can be misleading despite poor PPV.

To make this concrete, consider a condition with <1% prevalence. A model may achieve an AUC of 0.90 yet still have a PPV of only a few percent at clinically practical thresholds, meaning that the vast majority of “positive” predictions would be false alarms. Without PPV, NPV, the false-positive burden at fixed sensitivities, and appropriate precision–recall evaluation, it is impossible to determine whether the reported performance improvements or the conditional relationships inferred by the model translate into meaningful diagnostic value.

Response: We have introduced PPV and NPV along with a number of metrics in Table 4, using a second threshold corresponding to 90% sensitivity in the predictions as the reviewer pointed out the practical utility in measuring false alarms. All models appear to handle negative outcomes particularly well, and mainly differ in the PPV. We noticed that models that involved some form of joint modelling produced better PPVs, as we can see in Table 4 that the multitask neural network and multi-output random

forest produced significantly higher PPVs than any single-task/separate modelling. Our Bayesian multitask models still remain amongst the best performing by a small but statistically significant margin. For precision and recall, we describe what was carried out in the previous response: AUC-PR represents the average precision across all recall values and is evaluated on the raw probabilities; the AUC and AUC-PR curves are presented in Fig. S13. For exact precision and recall, which require converting predicted probabilities to binary, the F1 score was used as the objective to select the threshold as it provides a balance between precision and recall. We then considered both the binary F1 focusing on the positives, and the macro F1 to provide a balance between the precision on the positive and negative outcomes.

Comment:

4. Graph-derived conditional dependencies must be explicitly validated and interpreted

While the Discussion briefly relates some inferred condition relationships to known hepatosplenic pathways, this evaluation remains qualitative. Given that the graph-convolution component is central to the model’s interpretability claims, the manuscript should more explicitly and systematically analyze whether the conditional relationships inferred by the model align with established clinical mechanisms, such as known sequences of fibrosis progression, portal hypertension development, and splenic involvement. It would be valuable for the authors to indicate where the learned dependencies support existing pathophysiological understanding and where they diverge. Without such analysis, it is difficult to determine whether the smoothing imposed by the graph enhances biological realism or primarily reflects statistical correlation structure.

Response: Thank you for highlighting the importance of graph validation. We first added Table S9, showing pairs of nodes with the strongest probabilities in the graph convolution matrix, which also represent the strongest inter-dependencies in the graph. A paragraph of the clinical analysis on these edges has been added to the Discussion, where we have interpreted the biological correctness of Table S9, discussing a combination of biological complications, disease progression, and data acquisition as compared to the existing biological knowledge in the literature though no relationships were experimentally validated. The graph is learned from data and therefore inherently reflects the statistical relationships between the conditions. Efforts were made in using higher order distributional statistical measures to capture the most important relationship to best represent true relationships. The work in [10] does go into case analysis, visualizing a number of subgraphs to allow close examination of individual conditions and their related morbidities.

Comment:

5. Evidence for generalizability would strengthen the contribution

While external validation datasets for schistosomiasis are scarce, a geographically stratified validation (training on two districts and testing on the third) would provide valuable insight into model transportability. If this is not feasible, some robustness analysis (bootstrapping, district-weighted resampling, or sensitivity to graph structure) would still enhance the reliability of the

findings.

Response:

Thank you for the suggestions for new experiments. We have conducted these new experiments and added to them to the paper. First, we evaluated our Bayesian multitask model on district-based data splitting, training on two district and predicting on the third, which we present in the new Table S8. Given that district was a significant covariate, we expect the population from each district to have strong shared risk factors and there to be obvious district differences. It also should be noted that SchistoTrack was designed to include purposely different districts in terms of risk factors and prevalence so it is unclear whether a district validation shows generalisability or whether it is better to keep a model trained on diverse districts then let it see other settings that are contained in the diversity it is trained on. As expected, we find that splitting by district influences predictive performances. The result in Table S8 suggests mixed trends but generally Mayuge is the best predicted district with the highest average AUC-PR and PPV. But, this performance may be due to having most of the cases (highest condition prevalences) in the other two (Western districts) so for the metrics that considered imbalance there were most positive cases to be seen during training. These results are now included in the section "Multimorbidity prediction" and in the a Discussion paragraph.

We have also conducted graph perturbation to measure the variability of the influence probabilities from the graph convolutional matrix. We tried random addition of edges, and addition and removal; when edges are added, pairs of nodes are randomly selected and given weights that are bootstrap sampled from the list of existing edges. We showed what happens to the convolution matrix weights around gastro-oesophageal varices with 5%, 10%, and 20% edge removal, when both adding and removing are performed and the total number of edges remains the same. This experiment was ran a total of 100 times, and we present these in newly added Fig. S20 - S22. The graph convolution was very robust to graph perturbation especially at low percentages (5% & 10%) with very little variability to the convolution weights, while the unperturbed weights nearly always remained within the inter-quartile range from perturbation even at 20%. Thus, we concluded the graph was robust to noise, and this point was added to the Discussion paragraphs on the validity of the graph.

Comment:

6. Data availability and reproducibility

While I understand the ethical and privacy constraints around participant-level data, it would be helpful for the authors to clarify whether any form of anonymized, de-identified, or synthetic dataset could be shared to facilitate reproducibility. If full individual-level data cannot be released due to ethics board restrictions or data governance rules, the authors might still consider providing a minimally de-identified feature matrix (removing geographic, demographic, and other potentially identifying variables) or a synthetic dataset generated to match the statistical properties of the original cohort. Either option would enable readers to reproduce the modeling pipeline and evaluate the proposed method more effectively.

Response: For testing purposes, we now provide generated data by random sampling. For continuous covariates, we sampled from normal distributions with means and stan-

dard deviations equal to each of the variables (the original covariates had log transforms applied if they became closer to normal). For binary covariates, we sample from a Bernoulli using the mean of the variables. We also did this for each of the outcomes, using the prevalence of each condition. We re-uploaded these source files to allow the code to run, however we are unable to reproduce the results of the paper because this would involve generating data in a way that mimics the original data, which would be challenging especially for multiple outcomes and beyond the scope of this study—whereas the main aim of providing the anonymised code is to enable other studies to test our models on new datasets.

Comment:

Figure 3, including its schematic and legend, should be more explicative. This figure introduces important components of the model but currently requires substantial inference from the reader. Additional labels or a clearer narrative description would improve clarity.

Response: To make the pipeline figure clearer without reading the methods, we have added more a detailed description in the caption explaining what each panel does. Given the number of interpretive elements to the model, we completely agree that this needed more explanation to somewhat stand alone without reading all of the methods.

Remark on code availability:

I examined the shared materials (README.md and pyro_mcmc.py). Although the core Bayesian multitask model is implemented, the current repository does not allow readers to reproduce the results reported in the manuscript. The main script depends on a non-included module (data_loader) and on participant-level data that are not provided, so the pipeline cannot be executed end to end. The example at the bottom of the script does not run successfully either. It attempts to compute AUROC values but references a variable (test_id) that is undefined when no train–test split is created, meaning that the example cannot produce the metrics shown in the paper.

The underlying probabilistic model appears correctly implemented, but without a working example, without any form of anonymized or synthetic dataset, and without an evaluation pipeline that mirrors the manuscript, the results are not currently reproducible. To meet the reproducibility expectations of the journal, the authors should provide a functional example using either anonymized or synthetic data, ensure that AUROC values and other metrics can actually be computed and saved, and expand the evaluation to include imbalance-aware metrics. A more complete README and more robust covariate handling would also improve clarity and usability.

Response: The imported module “dataloader” contained information about our internal cloud structure and participants which was why it could not be shared. Its purpose was to load the condition outcomes, the rest of the “pyro_mcmc.py” script showed loading of the covariates, which were then joined with the outcomes. We wanted to show that the variables in the code corresponded to the ones described in the paper. The variables “test_id” and “train_id” were global variables that populate the session once you run “run_mcmc” or “run_mcmc_id”, we have changed this to explicitly defining them in the example usage. We now share the revised code that is much more reduced

with all data preparation removed, and will simply import the synthetic files. Note, the synthetic data does not replicate the actual data but is provided to allow the code to run smoothly for anyone learning to use our models. We cannot share the individual data or properties of it that can be relearned to identify communities due to our data protection impact assessment, ethics approvals, and ongoing nature of the cohort.

References

1. Garriga, R. *et al.* Machine learning model to predict mental health crises from electronic health records. *Nature medicine* **28**, 1240–1248 (2022).

Reviewer 3

This study deploys an interpretable Bayesian multitask learning framework to jointly model 45 hepatosplenic ultrasound outcomes in a rural Ugandan cohort. By combining clinical tests + sociodemographic + access/location covariates with a graph-convolution over an outcome-dependency network and continuous inclusion probabilities for covariates, the authors both (i) improve discrimination—especially for rare, severe outcomes—and (ii) surface clinically plausible risk patterns (older age↑; low Hb↑; female sex protective; higher risk among fishers). The work is methodologically thoughtful and practically relevant to triage in resource-limited settings. So it is suggested to make some revisions with following comments:

Response: We appreciate the positive assessment of this work from the reviewer, and are particularly thankful for them recognizing the strength of the methodology and the practical application. We were happy to make clarifications and the revisions suggested by the reviewer.

Comment:

1. Primary estimation & interpretability.

Most substantive findings in the text derive from per-outcome logistic regressions (ORs), while the proposed multitask model is judged mainly by AUC. If interpretability is central, the authors should report posterior log-odds (converted to ORs) with 95% HPD intervals for key covariates within the multitask model (e.g., $f_\alpha(G)W^\top \text{diag}(\Gamma)$ mapped to ORs after reversing standardization). Please also explain how Γ (continuous inclusion) relates to effect sizes and whether high- Γ variables consistently exhibit larger posterior odds shifts across tasks.

Response: We agree with the reviewer for pointing out that more interpretive statistics should be highlighted and more attention drawn to when these materials are provided in the supplement.

Firstly, we showed the posterior distributions of the odds ratios (posterior log-odds) for gastro-oesophageal varices, presented in supplementary Fig. S2. These are plotted on the log-scale and the 95% HPD intervals are also marked out for each covariate. All reported median and credible intervals in the results are based on these histograms. A set of 28 plots exist for each of the 45 conditions (so $28 \times 45 = 1260$ plots); such a large number of plots would have been difficult to include in a manuscript. As an example, we only displayed these plots for one condition (gastro-oesophageal varices) that we focused our analysis on. We also interpreted these plots in our results and discussion. We picked out the significant variables in age and (log) Hb concentration, with the median posterior odds ratios indicating 4% increase per year of age, and 22% increase per 10% decrease in log-transformed Hb concentration. In Table 3, all odds ratios are also presented with 95% HPD intervals. One difference is the significance of the covariates are determined via q -values from corrected p -values from likelihood ratio tests, instead of being determined by the HPD intervals containing odds ratio 1. The corrected q -values are a stricter measure to reduce false-positives and we can see that all significant variables had HPD intervals not including odds ratio 1. All odds ratios, both in the plots and tables, are

presented after reverse standardization and detailed this in the section "Risk factor association" in Methods.

The inclusion probabilities Γ alone do not determine the effect size, but do exhibit strong correlations with the size of odds ratios across all conditions. Their function is to spread information about the utility of the covariates. If a covariate is a very good predictor for a number of conditions, then it would lead to high γ , which will increase the influence of that covariate on the rest of the outcomes. As a result, the effect size of the covariate could be increased on outcomes that previously did not have a strong relationship. However the effect size is dependent on all terms in $f_\alpha(G)\mathbf{W}^\top \text{diag}(\Gamma)$, not just Γ , and each covariates' effect size can still be reduced by the \mathbf{W} terms.

Comment:

2. Validation design (80/20 and beyond).

Validation design should also be expanded and clearly described. If the data are split 80/20, please report repeated stratified 80/20 evaluation over many resamples; additionally, include a geographical transport test (leave-one-district-out) and, if feasible, a temporal validation to probe stability over time. For rare outcomes, present PR-AUC, F1-macro, and confusion matrices at clinically meaningful thresholds. Because clinical deployment hinges on threshold choices, decision-curve analysis would add practical value by quantifying net benefit across plausible risk thresholds.

Response: We apologize for the lack of clarity on the data splitting. The results are presented in two aspects, first, for explainability purposes, we obtained all statistics using the full 100% of the data which Table 3, S1 - S6, and Fig. 2, S1, - S4, and S6 - S11 were derived from; any report of medians and credible intervals were derived from 100% of the data. We now make this information clear (what data were used to obtain the results) in the caption of each table of figure. In the last section of the results, we reported testing the predictive ability of the model, and here we split the data into training and testing sets. We performed random splits of 50:50 for training and testing, and we resampled this way 10 times to produce standard deviations on the test AUCs. We chose 50:50 split because for rare conditions this will lead to more test sets that contain at least one positive and can be evaluated (note the model was able to learn from training data even if only one class existed). Fig. 3 and S5 show predictive performances and are therefore based on this method of splitting. We also clarify in the figure captions that this split was used.

While we understand temporal analysis will add significant insights to hepatosplenic multimorbidity, we are not able to complete this here due to our study being cross-sectional and only considered one year of data. Modelling of temporal patterns are currently ongoing within the cohort. And, our discussion highlights the importance of prospective designs in future work (in the Limitations section).

For the geographical prediction test, we have now evaluated our Bayesian multitask model on leave-one-out based on districts, we present this in the new Table S8. Given that district was a significant covariate, we expect the population from each district to have strong shared risk factors and there to be obvious district differences. It also should be noted that SchistoTrack was designed to include purposely different districts in terms of risk factors and prevalence so it is unclear whether a district validation

shows generalisability or whether it is better to keep a model trained on diverse districts then let it see other settings that are contained in the diversity it is trained on. As expected, we find that splitting by district influences predictive performances. The results suggest mixed trends but generally Mayuge is the best predicted district with the highest average AUC-PR and PPV. But, this performance may be due to having most of the cases (highest condition prevalences) in the other two (Western districts) so for the metrics that considered imbalance there were most positive cases to be seen during training.

We understand the need for imbalance-aware metrics, and have since evaluated our models on a list of metrics including AUC-PR, binary and macro F1, as well as precision, recall, and positive and negative predicted values (PPV & NPV). Importantly, we have changed our main predictive analysis to comparisons between AUC-PR and prevalence, replacing the original AUC comparisons. For binary based metrics, we have used two methods of choosing the threshold: first, the threshold leading to the highest F1 score, which we then used to evaluate the precision and recall of the model, as F1 provides a balance of the model precision and recall. We did this for both the binary F1 focusing on just the positive outcome, and the macro F1 which provides a balance between the positive and negative classes. For PPV and NPV, we set a second cut-off that leads to 90% sensitivity. Being a clinically informative metric, PPV then informs the likelihood of the patients truly having the condition having been predicted positive.

Comment:

3. Multimorbidity patterns (go beyond “better AUC”).

Since multimorbidity modeling is a stated novelty, I encourage the authors to move beyond “better AUC” and explicitly characterize comorbidity patterns induced by the model. Identify covariate scenarios that maximize predicted multimorbidity counts (e.g., low Hb combined with older age and longer distance to facilities); present patient-level archetypes by clustering predicted outcome vectors; and summarize which outcomes “donate” the most information to others via the graph by reporting row-wise weights from the graph convolution. Interpreting these pathways in light of hepatosplenic pathophysiology would substantiate the claim that the model surfaces clinically actionable “danger signals.”

Response: We thank the reviewer for suggesting the many different ways to interpret multimorbidity patterns and have expanded this element of the paper. First, we have added Discussion points on the results on overall multimorbidity from the last paragraph of "Risk factors for hepatosplenic conditions" section. The risk factors of overall multimorbidity were used to predict the probabilities of having all conditions, this means the sum of the probabilities represented the expected number of conditions, which informs the morbidity count. Therefore, we can determine the scenarios that would maximize expected morbidity count based on covariate significance and directions of the associations to "Overall" in Table S2; this includes low Hb concentration, older age, and four other factors. We also introduced Fig. S23 where we computed the probabilities of having every condition if a patient were to have this scenario of the covariates that maximizes morbidity counts, the setting is as follows: positively associated covariates (e.g. age) were assigned the 99th percentile value from our data, negatively associated covariates (e.g. Hb concentration) were assigned the 1st percentile value, insignificant

covariates were assigned the 50th percentile (median).

To measure the pairwise interactions between the outcomes in the model, we introduced Table S9 to document the highest weights between node pairs from the graph convolution matrix, representing the outcomes that donate the most information to each other. The interpretation of the biological correctness is based on the pairs highlighted in this table, and a paragraph is added in the Discussion which we would like to direct the reviewer to.

Because of the nature of the problem tackled in this study, we like to clarify that clustering can be performed on the outcomes or the participants, but there is not a clear connection between the two. Therefore, we cannot comment on within-patient archetypes based on clusters of conditions. We performed Louvain community detection on the graph to produce clusters of conditions and presented these in Fig. S24. We found a central cluster was particularly consistent with our analysis as it grouped gastro-oesophageal varices with all fibrosis patterns C - F, confirming its relation with periportal fibrosis. Other clusters also reflect biological patterns, with there existing a cluster containing all forms of enlargements on both liver lobes, while shrunkenness/restrictedness of both liver lobes were clustered together with various shrunkenness in the spleen and portal vein.

Comment:

Finally, robustness and reporting can be strengthened. Please conduct graph sensitivity analyses by perturbing edge weights/thresholds or re-learning the graph and assessing stability of performance and danger-signal rankings; perform prior sensitivity for the graph-strength and variance components; specify missing-data handling; and document prevalence and any class-imbalance strategies.

Response: We have firstly conducted graph perturbation and measured the variability of the influence probabilities from the graph convolutional matrix. We tried random addition of edges, and addition and removal; when edges are added, pairs of nodes are randomly selected and given weights that are bootstrap sampled from the list of existing edges. We set the proportions of edges to add/remove at 5%, 10%, and 20%, when both adding and removing are performed the total number of edges remain the same. This was run a total of 100 times. We measured the range of values from the graph convolution around gastro-oesophageal varices, which we presented in newly added Fig. S20 - S22. The graph convolution function we used was very robust to graph perturbation especially at low percentages (5% & 10%) there were very little variability to the convolution weights, while the original unperturbed convolution weights nearly always remained within the inter-quartile range even at 20%. This finding is integrated in the Discussion along with the interpretation on the strongest graph connections to support the validation of the multimorbidity graph.

For prior selection, we have tested various options before settling on the priors specified in our paper. For the α parameter in the graph convolution, we tested Gamma priors $\Gamma(\alpha, \beta)$ with the mode values at 1, 3, and 5 (mean = α/β), the performance was similar for mode = 1 and 3, but worse for mode = 5, suggesting it should be a small value, and a mode of 1 was used in our final model. Additionally, the inclusion probabilities γ_i was the most robust to priors with variously shaped Beta distributions tested that

all led to almost identical posteriors. For the regression coefficients, normal and t priors were tested and did not show noticeable difference, but the performance was sensitive to the variance/scale terms of the normal/ t distribution. This led to us selecting the normal and putting a prior on the variance to allow it to vary. We have added these explanations to the "Prior selection" section in Methods, and displayed the posteriors using an alternative set of priors in Fig. S3 (see caption for priors used), to show that as long as the prior selection were reasonable for each model component the results are extremely robust.

Participants are selected for this study only if they have all required data. Of the participants examined with point-of-care ultrasound, only 2.14% (69/3224) had missing ultrasound data or covariates, therefore we decided it was reasonable to exclude them from the dataset. This is detailed in Participants selection section of Methods, and a participants flowchart in Supplementary which details the number of missingness at each stage of the data collection.

We have documented the prevalence of every outcome condition in Table 1, the number of participants having each condition is listed with the prevalence in bracket. There is a strong imbalance especially for a number of severe conditions, and this partly motivated the model architecture and framework. Firstly, being a fully Bayesian model, the use of priors act as regularization on the model parameters. Bayesian inference also incorporates the distributions of the parameters in the prediction through marginalization and Bayesian averaging, instead of using any point estimate that could potentially lead to overfitting. Architecture-wise, the multitask framework induces parameter sharing between the models for each outcome, thereby all parameters are exposed to more outcomes, and in turn more positive outcomes. Furthermore, we have introduced a number of imbalance-aware metrics including AUC-PR, F1, precision, recall, and PPV to provide clearer evaluations of the predictions.